# ON THE SOFT-SUBNETWORK FOR
# FEW-SHOT CLASS INCREMENTAL LEARNING

**Haeyong Kang, Jaehong Yoon, Sultan Rizky Madjid, Sung Ju Hwang, and Chang D. Yoo**[*]
Korea Advanced Institute of Science and Technology (KAIST)
291 Daehak-ro, Yuseong-gu, Daejeon
{`haeyong.kang,jaehong.yoon,suulkyy,sjhwang82,cd_yoo`}@kaist.ac.kr

## ABSTRACT

Inspired by *Regularized Lottery Ticket Hypothesis*, which states that competitive smooth (non-binary) subnetworks exist within a dense network, we propose a few-shot class-incremental learning method referred to as *Soft-SubNetworks (SoftNet)*. Our objective is to learn a sequence of sessions incrementally, where each session only includes a few training instances per class while preserving the knowledge of the previously learned ones. SoftNet jointly learns the model weights and adaptive non-binary soft masks at a base training session in which each mask consists of the major and minor subnetwork; the former aims to minimize catastrophic forgetting during training, and the latter aims to avoid overfitting to a few samples in each new training session. We provide comprehensive empirical validations demonstrating that our SoftNet effectively tackles the few-shot incremental learning problem by surpassing the performance of state-of-the-art baselines over benchmark datasets. The public code is available at https://github.com/ihaeyong/SoftNet-FSCIL.

## 1 INTRODUCTION

Lifelong Learning, or Continual Learning, is a learning paradigm to expand knowledge and skills through sequential training of multiple tasks (Thrun, 1995). According to the accessibility of task identity during training and inference, the community often categorizes the field into specific problems, such as task-incremental (Pfülb and Gepperth, 2019; Delange et al., 2021; Yoon et al., 2020; Kang et al., 2022), class-incremental (Chaudhry et al., 2018; Kuzborskij et al., 2013; Li and Hoiem, 2017; Rebuffi et al., 2017; Kemker and Kanan, 2017; Castro et al., 2018; Hou et al., 2019; Wu et al., 2019), and task-free continual learning (Aljundi et al., 2019; Jin et al., 2021; Pham et al., 2022; Harrison et al., 2020). While the standard scenarios for continual learning assume a sufficiently large number of instances per task, a lifelong learner for real-world applications often suffers from insufficient training instances for each problem to solve. This paper aims to tackle the issue of limited training instances for practical Class-Incremental Learning (CIL), referred to as Few-Shot CIL (FSCIL) (Ren et al., 2019; Chen and Lee, 2020; Tao et al., 2020; Zhang et al., 2021; Cheraghian et al., 2021; Shi et al., 2021).

However, there are two critical challenges in solving FSCIL problems: *catastrophic forgetting* and *overfitting*. Catastrophic forgetting (Goodfellow et al., 2013; Kirkpatrick et al., 2017) or Catastrophic Interference McCloskey and Cohen (1989) is a phenomenon in which a continual learner loses the previously learned task knowledge by updating the weights to adapt to new tasks, resulting in significant performance degeneration on previous tasks. Such undesired knowledge drift is irreversible since the scenario does not allow the model to revisit past task data. Recent works propose to mitigate catastrophic forgetting for class-incremental learning, often categorized in multiple directions, such as *constraint-based* (Rebuffi et al., 2017; Castro et al., 2018; Hou et al., 2018; 2019; Wu et al., 2019), *memory-based* (Rebuffi et al., 2017; Chen and Lee, 2020; Mazumder et al., 2021; Shi et al., 2021), and *architecture-based methods* (Mazumder et al., 2021; Serra et al., 2018; Mallya and Lazebnik, 2018; Kang et al., 2022). However, we note that catastrophic forgetting becomes further challenging

---

[*]Corresponding Author.

in FSCIL. Due to the small amount of training data for new tasks, the model tends to severely **overfit to new classes** and quickly forget old classes, deteriorating the model performance.

Meanwhile, several works address overfitting issues for continual learning from various perspectives. NCM (Hou et al., 2019) and BiC (Wu et al., 2019) highlight the prediction bias problem during sequential training that the models are prone to predict the data to classes in recently trained tasks. OCS (Yoon et al., 2022) tackles the class imbalance problems for rehearsal-based continual learning, where the number of instances at each class varies per task so that the model would perform biased training on dominant classes. Nevertheless, these works do not consider the overfitting issues caused by training a sequence of few-shot tasks. FSLL (Mazumder et al., 2021) tackles overfitting for few-shot CIL by partially-splitting model parameters for different sessions through multiple substeps of iterative reidentification and weight selection. However, it led to computationally inefficient.

To deploy a practical few-shot CIL model, we propose a simple yet efficient method named *SoftNet*, effectively alleviating catastrophic forgetting and overfitting. Motivated by *Lottery Ticket Hypothesis* (Frankle and Carbin, 2019), which hypothesizes the existence of competitive subnetworks (winning tickets) within the randomly initialized dense neural network, we suggest a new paradigm for Few-shot CIL, named *Regularized Lottery Ticket Hypothesis*:

**Regularized Lottery Ticket Hypothesis (RLTH).** *A randomly-initialized dense neural network contains a regularized subnetwork that can retain the prior class knowledge while providing room to learn the new class knowledge through isolated training of the subnetwork.*

Based on RLTH, we propose a method, referred to as **Soft**-Sub**Net**works (**SoftNet**), illustrated in Figure 1. First, SoftNet jointly learns the randomly initialized dense model (Figure 1 (a)) and soft mask $m \in [0,1]^{|\theta|}$ pertaining to Soft-subnetwork (Figure 1 (b)) on the base session training; the soft mask consists of the major part of the model parameters $m = 1$ and the minor ones $m < 1$ where $m = 1$ is obtained by the top-$c\%$ of model parameters and $m < 1$ is obtained by the remaining ones $(100- \text{top-}c\%)$ sampled from the uniform distribution. Then, we freeze the major part of pre-trained subnetwork weights for maintaining prior class knowledge and update the only minor part of weights for the novel class knowledge (Figure 1 (c)).

We summarize our key contributions as follows:

- This paper presents a new masking-based method, *Soft-SubNetwork (SoftNet)*, that tackles two critical challenges in the few-shot class incremental learning (FSCIL), known as catastrophic forgetting and overfitting.

- Our SoftNet trains two different types of non-binary masks (subnetworks) for solving FSCIL, preventing the continual learner from forgetting previous sessions and overfitting simultaneously.

- We conduct a comprehensive empirical study on SoftNet with multiple class incremental learning methods. Our method significantly outperforms strong baselines on benchmark tasks for FSCIL problems.

## 2 RELATED WORK

**Catastrophic Forgetting.** Many recent works have made remarkable progress in tackling the challenges of catastrophic forgetting in lifelong learning. To be specific, Architecture-based approaches (Mallya et al., 2018; Serrà et al., 2018; Li et al., 2019) utilize an additional capacity to expand (Xu and Zhu, 2018; Yoon et al., 2018) or isolate (Rusu et al., 2016) model parameters, thereby avoiding knowledge interference during continual learning; SupSup (Wortsman et al., 2020) allocates model parameters dedicated to different tasks. Very recently, Chen et al. (2021); Kang et al. (2022) shows the existence of a sparse subnetwork, called winning tickets, that performs well on all tasks during continual learning. However, many subnetwork-based approaches are incompatible with the FSCIL setting since performing task inference under data imbalances is challenging. FSLL (Mazumder et al., 2021) aims to search session-specific subnetworks while preserving weights for previous sessions for incremental few-shot learning. However, the expansion process comprises another series of retraining and pruning steps, requiring excessive training time and computational costs. On the contrary, our proposed method, SoftNet, jointly learns the model and task-adaptive

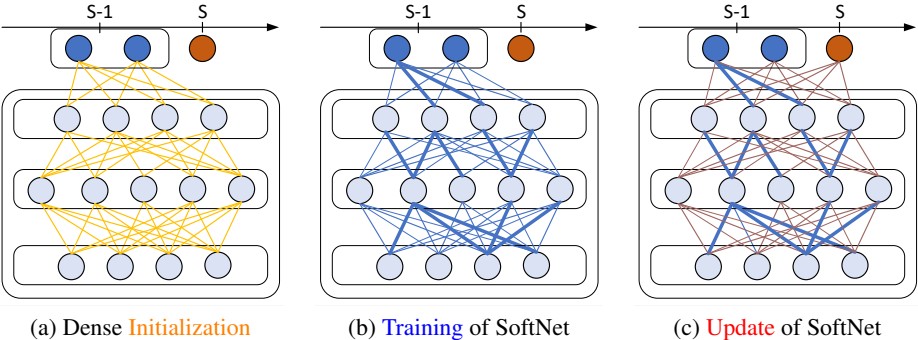

(a) Dense Initialization      (b) Training of SoftNet      (c) Update of SoftNet

Figure 1: **Incremental Soft-Subnetwork (SoftNet):** (a) Dense Neural Network is randomly initialized for the base session (S-1) training (b) SoftNet is trained by major subnetwork $\boldsymbol{m}_{major} = 1$ (thick solid line) and minor $\boldsymbol{m}_{minor} \sim U(0, 1)$, and (c) SoftNet updates only a few minor weights (thin solid line) for new sessions (S).

smooth (i.e., non-binary) masks of the subnetwork associated with the base session while selecting an essential subset of the model weights for the upcoming session. Furthermore, smooth masks behave like regularizers that prevent overfitting when learning new classes.

**Soft-subnetwork.** Recent works with context-dependent gating of sub-spaces (He and Jaeger, 2018), parameters (Mallya and Lazebnik, 2018; He et al., 2019; Mazumder et al., 2021), or layers (Serra et al., 2018) of a single deep neural network demonstrated its effectiveness in addressing catastrophic forgetting during continual learning. Masse et al. (2018) combines context-dependent gating with the constraints preventing significant changes in model weights, such as SI (Zenke et al., 2017) and EWC (Kirkpatrick et al., 2017), achieving further performance increases than using them alone. A flat minima could also be considered as acquiring sub-spaces. Previous works have shown that a flat minimizer is more robust to random perturbations (Hinton and Van Camp, 1993; Hochreiter and Schmidhuber, 1994; Jiang et al., 2019). Recently, Shi et al. (2021) showed that obtaining flat loss minima in the base session, which stands for the first task session with sufficient training instances, is necessary to alleviate catastrophic forgetting in FSCIL. To minimize forgetting, they updated the model weights on the obtained flat loss contour. In our work, by selecting sub-networks (Frankle and Carbin, 2019; Zhou et al., 2019; Wortsman et al., 2019; Ramanujan et al., 2020; Kang et al., 2022; Chijiwa et al., 2022) and optimizing the sub-network parameters in a sub-space, we propose a new method to preserve learned knowledge from a base session on a major subnetwork and learn new sessions through regularized minor subnetworks.

## 3 SOFT-SUBNETWORK FOR FEW-SHOT CLASS INCREMENTAL LEARNING

### 3.1 PROBLEM STATEMENTS

Various works have tried to mitigate catastrophic forgetting problems in class incremental learning using knowledge distillation, revisiting a subset of prior samples, or isolating essential model parameters to retain prior class knowledge even after the model loses accessibility to them. However, as a few-shot class incremental learning scenario regards following tasks/sessions containing a small amount of training data, the model tends to severely overfit to new classes, making it difficult to fine-tune the previously trained model on a few samples. In addition, the fine-tuning process often leads to the catastrophic forgetting of base class knowledge. As a result, regularization is indispensable in the models to avoid forgetting and prevent the model from overfitting to new class samples by updating only the selected parameters for learning in the new session.

**Few-shot Class Incremental Learning** (FSCIL) aims to learn new sessions with only a few examples continually. A FSCIL model learns a sequence of $T$ training sessions $\{\mathcal{D}^1, \cdots, \mathcal{D}^T\}$, where $\mathcal{D}^t = \{z_i^t = (\boldsymbol{x}_i^t, y_i^t)\}_i^{n_t}$ is the training data of session $t$ and $\boldsymbol{x}_i^t$ is an example of class $y_i^t \in \mathcal{O}^t$. In FSCIL, the base session $\mathcal{D}^1$ usually contains a large number of classes with sufficient training data for each class. In contrast, the subsequent sessions ($t \geq 2$) will only contain a small number of classes with a few training samples per class, e.g., the $t^{\text{th}}$ session $\mathcal{D}^t$ is often presented as a *N*-way *K*-shot task. In each training session $t$, the model can access only the training data $\mathcal{D}^t$ and a few examples stored

in the previous session. When the training of session $t$ is completed, we evaluate the model on test samples from all classes $\mathcal{O} = \bigcup_{i=1}^{t} \mathcal{O}^i$, where $\mathcal{O}^i \bigcap \mathcal{O}^{j \neq i} = \emptyset$ for $\forall i, j \leq T$.

Consider a supervised learning setup where the $T$ sessions arrive in a lifelong learner $f(\cdot; \boldsymbol{\theta})$ parameterized by the model weights $\boldsymbol{\theta}$ in sequential order. A few-shot class incremental learning scenario aims to learn the classes in a sequence of sessions without catastrophic forgetting. In the training session $t$, the model solves the following optimization procedure:

$$\boldsymbol{\theta}^* = \underset{\boldsymbol{\theta}}{\text{minimize}} \frac{1}{n_t} \sum_{i=1}^{n_t} \mathcal{L}_t(f(\boldsymbol{x}_i^t; \boldsymbol{\theta}), y_i^t), \tag{1}$$

where $\mathcal{L}_t$ is a classification loss like cross-entropy, and $n_t$ is the number of instances for session $t$.

## 3.2 SUBNETWORK-BASED TRAINING FOR FEW-SHOT CLASS INCREMENTAL LEARNING

As lifelong learners often adopt over-parameterized dense neural networks to allow resource freedom for future classes or tasks, updating entire weights in neural networks for few-shot tasks is often not preferable and often yields the overfitting problem. To overcome the limitations in FSCIL, we focus on updating partial weights in neural networks when a new task arrives. The desired set of partial weights, named subnetwork, can achieve on-par or even better performance with the following motivations: (1) Lottery Ticket Hypothesis (Frankle and Carbin, 2019) shows the existence of a subnetwork that performs well as the dense network, and (2) The subnetwork significantly downsized from the dense network reduces the size of the expansion of the solver while providing extra capacity to learn new sessions or tasks.

We first suggest the objective referred to as HardNet as follows: given dense neural network parameters $\boldsymbol{\theta}$, the binary attention mask $\boldsymbol{m}_t^*$ describes the optimal subnetwork for session $t$ such that $|\boldsymbol{m}_t^*|$ is less than the dense model capacity $|\boldsymbol{\theta}|$. However, such binarized subnetworks $\boldsymbol{m}_t \in \{0, 1\}^{|\boldsymbol{\theta}|}$ cannot adjust the remaining parameters in a dense network for future sessions while solving past task problems cost- and memory efficiently. In FSCIL, the test accuracy of the base session drops significantly when it proceeds to learn sequential sessions since the subnetwork of $m = 1$ plays a crucial role in maintaining the base class knowledge. To this end, we propose a soft-subnetwork $\boldsymbol{m}_t \in [0, 1]^{|\boldsymbol{\theta}|}$ instead of the binarized subnetwork. It gives more flexibility to fine-tune a small part of the soft-subnetwork while fixing the rest to retain base class knowledge for FSCIL. As such, we find the soft-subnetwork through the following objective:

$$\boldsymbol{m}_t^* = \underset{\boldsymbol{m}_t \in [0,1]^{|\boldsymbol{\theta}|}}{\text{minimize}} \frac{1}{n_t} \sum_{i=1}^{n_t} \mathcal{L}_t \big( f(\boldsymbol{x}_i^t; \boldsymbol{\theta} \odot \boldsymbol{m}_t), y_i^t \big) - \mathcal{J} \tag{2}$$

$$\text{subject to } |\boldsymbol{m}_t| \leq c.$$

where session loss $\mathcal{J} = \mathcal{L}\big(f(\boldsymbol{x}_i^t; \boldsymbol{\theta}), y_i^t\big)$, the subnetwork sparsity $c \ll |\boldsymbol{\theta}|$ (used as the selected proportion % of model parameters in the following section), and $\odot$ represents an element-wise product. In the following section, we describe how to obtain the soft-subnetwork $\boldsymbol{m}_t^*$ using the magnitude-based criterion (RLTH) while minimizing session loss jointly.

## 3.3 OBTAINING SOFT-SUBNETWORKS VIA COMPLEMENTARY WINNING TICKETS

Let each weight be associated with a learnable parameter we call *weight score $s$*, which numerically determines the importance of the associated weight. In other words, we declare a weight with a higher score as more important. At first, we find a subnetwork $\boldsymbol{\theta}^* = \boldsymbol{\theta} \odot \boldsymbol{m}_t^*$ of the dense neural network and then assign it as a solver of the current session $t$. The subnetworks associated with each session jointly learn the model weight $\boldsymbol{\theta}$ and binary mask $\boldsymbol{m}_t$. Given an objective $\mathcal{L}_t$, we optimize $\boldsymbol{\theta}$ as follows:

$$\boldsymbol{\theta}^*, \boldsymbol{m}_t^* = \underset{\boldsymbol{\theta}, \boldsymbol{s}}{\text{minimize}} \, \mathcal{L}_t(\boldsymbol{\theta} \odot \boldsymbol{m}_t; \mathcal{D}_t). \tag{3}$$

where $\boldsymbol{m}_t$ is obtained by applying an indicator function $\mathbb{1}_c$ on weight scores $\boldsymbol{s}$. Note $\mathbb{1}_c(s) = 1$ if $s$ belongs to top-$c$% scores and 0 otherwise.

In the optimization process for FSCIL, however, we consider two main problems: (1) Catastrophic forgetting: updating all $\boldsymbol{\theta} \odot \boldsymbol{m}_{t-1}$ when training for new sessions will cause interference with

the weights allocated for previous tasks; thus, we need to freeze all previously learned parameters $\boldsymbol{\theta} \odot \boldsymbol{m}_{t-1}$; (2) Overfitting: the subnetwork also encounters overfitting issues when training an incremental task on a few samples, as such, we need to update a few parameters irrelevant to previous task knowledge., i.e., $\boldsymbol{\theta} \odot (\mathbf{1} - \boldsymbol{m}_{t-1})$.

To acquire the optimal subnetworks that alleviate the two issues, we define a soft-subnetwork by dividing the dense neural network into two parts-one is the major subnetwork $\boldsymbol{m}_{\text{major}}$, and another is the minor subnetwork $\boldsymbol{m}_{\text{minor}}$. The defined soft-subnetwork follows as:

$$\boldsymbol{m}_{\text{soft}} = \boldsymbol{m}_{\text{major}} \oplus \boldsymbol{m}_{\text{minor}}, \tag{4}$$

where $\boldsymbol{m}_{\text{major}}$ is a binary mask and $\boldsymbol{m}_{\text{minor}} \sim U(0,1)$ and $\oplus$ represents an element-wise summation. As such, a soft-mask is given as $\boldsymbol{m}_t^* \in [0,1]^{|\boldsymbol{\theta}|}$ in Eq.3. In the all-experimental FSCIL setting, $\boldsymbol{m}_{\text{major}}$ maintains the base task knowledge $t = 1$ while $\boldsymbol{m}_{\text{minor}}$ acquires the novel task knowledge $t \geq 2$. Then, with base session learning rate $\alpha$, the $\boldsymbol{\theta}$ is updated as follows: $\boldsymbol{\theta} \leftarrow \boldsymbol{\theta} - \alpha \left( \frac{\partial \mathcal{L}}{\partial \boldsymbol{\theta}} \odot \boldsymbol{m}_{\text{soft}} \right)$ effectively regularize the weights of the subnetworks for incremental learning. The subnetworks are obtained by the indicator function that always has a gradient value of $\mathbf{0}$; therefore, updating the weight scores $\boldsymbol{s}$ with its loss gradient is impossible. To update the weight scores, we use Straight-through Estimator (Hinton, 2012; Bengio et al., 2013; Ramanujan et al., 2020) in the backward pass. Specifically, we ignore the derivatives of the indicator function and update the weight score $\boldsymbol{s} \leftarrow \boldsymbol{s} - \alpha \left( \frac{\partial \mathcal{L}}{\partial \boldsymbol{s}} \odot \boldsymbol{m}_{\text{soft}} \right)$, where $\boldsymbol{m}_{\text{soft}} = \mathbf{1}$ for exploring the optimal subnetwork for base session training. Our Soft-subnetwork optimizing procedure is summarized in Algorithm 1. Once a single soft-subnetwork $\boldsymbol{m}_{\text{soft}}$ is obtained in the base session, then we use the soft-subnetwork for the entire new sessions without updating.

---

**Algorithm 1** Soft-Subnetworks (SoftNet)

---

**input** $\{\mathcal{D}^t\}_{t=1}^{\mathcal{T}}$, model weights $\boldsymbol{\theta}$, and score weights $\boldsymbol{s}$, layer-wise capacity $c$
1:   // Training over base classes $t = 1$
2:   Randomly initialize $\boldsymbol{\theta}$ and $\boldsymbol{s}$.
3:   **for** epoch $e = 1, 2, \cdots$ **do**
4:       Obtain softmask $\boldsymbol{m}_{\text{soft}}$ of $\boldsymbol{m}_{major}$ and $\boldsymbol{m}_{minor} \sim U(0,1)$ at each layer
5:       **for** batch $\boldsymbol{b}_t \sim \mathcal{D}^t$ **do**
6:           Compute $\mathcal{L}_{base}\left(\boldsymbol{\theta} \odot \boldsymbol{m}_{\text{soft}}; \boldsymbol{b}_t\right)$ by Eq. 3
7:           $\boldsymbol{\theta} \leftarrow \boldsymbol{\theta} - \alpha \left( \frac{\partial \mathcal{L}}{\partial \boldsymbol{\theta}} \odot \boldsymbol{m}_{\text{soft}} \right)$
8:           $\boldsymbol{s} \leftarrow \boldsymbol{s} - \alpha \left( \frac{\partial \mathcal{L}}{\partial \boldsymbol{s}} \odot \boldsymbol{m}_{\text{soft}} \right)$
9:       **end for**
10:  **end for**
11:  // Incremental learning $t \geq 2$
12:  Combine the training data $\mathcal{D}^t$ and the exemplars saved in previous few-shot sessions
13:  **for** epoch $e = 1, 2, \cdots$ **do**
14:      **for** batch $\boldsymbol{b}_t \sim \mathcal{D}^t$ **do**
15:          Compute $\mathcal{L}_m\left(\boldsymbol{\theta} \odot \boldsymbol{m}_{\text{soft}}; \boldsymbol{b}_t\right)$ by Eq. 5
16:          $\boldsymbol{\theta} \leftarrow \boldsymbol{\theta} - \beta \left( \frac{\partial \mathcal{L}}{\partial \boldsymbol{\theta}} \odot \boldsymbol{m}_{minor} \right)$
17:      **end for**
18:  **end for**
**output** model parameters $\boldsymbol{\theta}$, $\boldsymbol{s}$, and $\boldsymbol{m}_{\text{soft}}$.

---

## 4 INCREMENTAL LEARNING FOR SOFT-SUBNETWORK

We now describe the overall procedure of our soft-pruning-based incremental learning/inference method, including the training phase with a normalized informative measurement in Section 4.1, as followed by the prior work (Shi et al., 2021), and the inference phase in Section 4.2.

### 4.1 INCREMENTAL SOFT-SUBNETWORK TRAINING

**Base Training** ($t = 1$). In the base learning session, we optimize the soft-subnetwork parameter $\boldsymbol{\theta}$ (including a fully-connected layer as a classifier) and weight score $\boldsymbol{s}$ with cross-entropy loss jointly using the training examples of $\mathcal{D}^1$.

**Incremental Training** ($t \geq 2$). In the incremental few-shot learning sessions ($t \geq 2$), leveraged by $\boldsymbol{\theta} \odot \boldsymbol{m}_{\text{soft}}$, we fine-tune few minor parameters $\boldsymbol{\theta} \odot \boldsymbol{m}_{\text{minor}}$ of the soft-subnetwork to learn new classes.

Table 1: Classification accuracy of ResNet18 on CIFAR-100 for 5-way 5-shot incremental learning. Underbar denotes the comparable results with FSLL (Mazumder et al., 2021). ∗ denotes the results reported from Shi et al. (2021).

| Method | sessions | | | | | | | | | The gap with cRT |
|---|---|---|---|---|---|---|---|---|---|---|
| | 1 | 2 | 3 | 4 | 5 | 6 | 7 | 8 | 9 | |
| cRT (Shi et al., 2021) | 65.18 | 63.89 | 60.20 | 57.23 | 53.71 | 50.39 | 48.77 | 47.29 | 45.28 | - |
| iCaRL (Rebuffi et al., 2017)∗ | 66.52 | 57.26 | 54.27 | 50.62 | 47.33 | 44.99 | 43.14 | 41.16 | 39.49 | -5.79 |
| Rebalance (Hou et al., 2019)∗ | 66.66 | 61.42 | 57.29 | 53.02 | 48.85 | 45.68 | 43.06 | 40.56 | 38.35 | -6.93 |
| FSLL (Mazumder et al., 2021)∗ | 65.18 | 56.24 | 54.55 | 51.61 | 49.11 | 47.27 | 45.35 | 43.95 | 42.22 | -3.08 |
| iCaRL (Rebuffi et al., 2017) | 64.10 | 53.28 | 41.69 | 34.13 | 27.93 | 25.06 | 20.41 | 15.48 | 13.73 | -31.55 |
| Rebalance (Hou et al., 2019) | 64.10 | 53.05 | 43.96 | 36.97 | 31.61 | 26.73 | 21.23 | 16.78 | 13.54 | -31.74 |
| TOPIC (Cheraghian et al., 2021) | 64.10 | 55.88 | 47.07 | 45.16 | 40.11 | 36.38 | 33.96 | 31.55 | 29.37 | -15.91 |
| F2M (Shi et al., 2021) | 64.71 | 62.05 | 59.01 | 55.58 | 52.55 | 49.96 | 48.08 | 46.28 | 44.67 | -0.61 |
| FSLL (Mazumder et al., 2021) | 64.10 | 55.85 | 51.71 | 48.59 | 45.34 | 43.25 | 41.52 | 39.81 | 38.16 | -7.12 |
| HardNet, $c = 50\%$ | 64.80 | 60.77 | 56.95 | 53.53 | 50.40 | 47.82 | 45.93 | 43.95 | 41.91 | -3.37 |
| HardNet, $c = 80\%$ | 69.65 | 64.60 | 60.59 | 56.93 | 53.60 | 50.80 | 48.69 | 46.69 | 44.63 | -0.65 |
| HardNet, $c = 99\%$ | 71.95 | 66.83 | 62.75 | 59.09 | 55.92 | 53.03 | 50.78 | 48.52 | 46.31 | +1.03 |
| SoftNet, $c = 50\%$ | 69.20 | 64.18 | 60.01 | 56.43 | 53.11 | 50.62 | 48.60 | 46.51 | 44.61 | -0.67 |
| SoftNet, $c = 80\%$ | 70.38 | 65.04 | 60.94 | 57.26 | 54.13 | 51.58 | 49.52 | 47.36 | 45.16 | -0.12 |
| SoftNet, $c = 99\%$ | **72.62** | **67.31** | **63.05** | **59.39** | **56.00** | **53.23** | **51.06** | **48.83** | **46.63** | **+1.35** |

Since $\boldsymbol{m}_{\mathrm{minor}} < \boldsymbol{1}$, the soft-subnetwork alleviates the overfitting of a few samples. Furthermore, instead of Euclidean distance (Shi et al., 2021), we employ a metric-based classification algorithm with cosine distance to finetune the few selected parameters. In some cases, Euclidean distance fails to give the real distances between representations, especially when two points with the same distance from prototypes do not fall in the same class. In contrast, representations with a low cosine distance are located in the same direction from the origin, providing a normalized informative measurement. We define the loss function as:

$$\mathcal{L}_m(z; \boldsymbol{\theta} \odot \boldsymbol{m}_{soft}) = -\sum_{z \in \mathcal{D}} \sum_{o \in \mathcal{O}} \mathbb{1}(y = o) \log \left( \frac{e^{-d(\boldsymbol{p}_o, f(\boldsymbol{x}; \boldsymbol{\theta} \odot \boldsymbol{m}_{soft}))}}{\sum_{o_k \in \mathcal{O}} e^{-d(\boldsymbol{p}_{o_k}, f(\boldsymbol{x}; \boldsymbol{\theta} \odot \boldsymbol{m}_{soft}))}} \right) \quad (5)$$

where $d(\cdot, \cdot)$ denotes cosine distance, $\boldsymbol{p}_o$ is the prototype of class $o$, $\mathcal{O} = \bigcup_{i=1}^{t} \mathcal{O}^i$ refers to all encountered classes, and $\mathcal{D} = \mathcal{D}^t \bigcup \mathcal{P}$ denotes the union of the current training data $\mathcal{D}^t$ and the exemplar set $\mathcal{P} = \{\boldsymbol{p}_2 \cdots, \boldsymbol{p}_{t-1}\}$, where $\mathcal{P}_{t_e} (2 \leq t_e < t)$ is the set of saved exemplars in session $t_e$. Note that the prototypes of new classes are computed by $\boldsymbol{p}_o = \frac{1}{N_o} \sum_i \mathbb{1}(y_i = o) f(\boldsymbol{x}_i; \boldsymbol{\theta} \odot \boldsymbol{m}_{soft})$ and those of base classes are saved in the base session, and $N_o$ denotes the number of the training images of class $o$. We also save the prototypes of all classes in $\mathcal{O}^t$ for later evaluation.

## 4.2 INFERENCE FOR INCREMENTAL SOFT-SUBNETWORK

In each session, the inference is also conducted by a simple nearest class mean (NCM) classification algorithm (Mensink et al., 2013; Shi et al., 2021) for fair comparisons. Specifically, all the training and test samples are mapped to the embedding space of the feature extractor $f$, and Euclidean distance $d_u(\cdot, \cdot)$ is used to measure the similarity between them. The classifier gives the $k$th prototype index $o_k^* = \arg\min_{o \in \mathcal{O}} d_u(f(\boldsymbol{x}; \boldsymbol{\theta} \odot \boldsymbol{m}_{soft}), \boldsymbol{p}_o)$ as output.

## 5 EXPERIMENTS

We introduce experimental setups in Section 5.1. Then, we empirically evaluate our soft-subnetworks for incremental few-shot learning and demonstrate its effectiveness through comparison with state-of-the-art methods and vanilla subnetworks in the following subsections.

## 5.1 EXPERIMENTAL SETUP

**Datasets.** To validate the effectiveness of the soft-subnetwork, we follow the standard FSCIL experimental setting. We randomly select 60 classes as the base class and the remaining 40 as new classes for CIFAR-100 and miniImageNet. In each incremental learning session, we construct 5-way 5-shot tasks by randomly picking five classes and sampling five training examples for each class.

Table 2: Classification accuracy of ResNet18 on miniImageNet for 5-way 5-shot incremental learning. Underbar denotes the comparable results with FSLL Mazumder et al. (2021). ∗ denotes the results reported from Shi et al. (2021).

| Method | sessions | | | | | | | | | The gap with cRT |
|---|---|---|---|---|---|---|---|---|---|---|
| | 1 | 2 | 3 | 4 | 5 | 6 | 7 | 8 | 9 | |
| cRT (Shi et al., 2021) | 67.30 | 64.15 | 60.59 | 57.32 | 54.22 | 51.43 | 48.92 | 46.78 | 44.85 | - |
| iCaRL (Rebuffi et al., 2017)* | 67.35 | 59.91 | 55.64 | 52.60 | 49.43 | 46.73 | 44.13 | 42.17 | 40.29 | -4.56 |
| Rebalance (Hou et al., 2019)* | 67.91 | 63.11 | 58.75 | 54.83 | 50.68 | 47.11 | 43.88 | 41.19 | 38.72 | -6.13 |
| FSLL (Mazumder et al., 2021)* | 67.30 | 59.81 | 57.26 | 54.57 | 52.05 | 49.42 | 46.95 | 44.94 | 42.87 | -1.11 |
| iCaRL (Rebuffi et al., 2017) | 61.31 | 46.32 | 42.94 | 37.63 | 30.49 | 24.00 | 20.89 | 18.80 | 17.21 | -27.64 |
| Rebalance (Hou et al., 2019) | 61.31 | 47.80 | 39.31 | 31.91 | 25.68 | 21.35 | 18.67 | 17.24 | 14.17 | -30.68 |
| TOPIC (Cheraghian et al., 2021) | 61.31 | 50.09 | 45.17 | 41.16 | 37.48 | 35.52 | 32.19 | 29.46 | 24.42 | -20.43 |
| IDLVQ-C (Chen and Lee, 2020) | 64.77 | 59.87 | 55.93 | 52.62 | 49.88 | 47.55 | 44.83 | 43.14 | 41.84 | -3.01 |
| F2M (Shi et al., 2021) | 67.28 | 63.80 | 60.38 | 57.06 | 54.08 | 51.39 | 48.82 | 46.58 | 44.65 | -0.20 |
| FSLL (Mazumder et al., 2021) | 66.48 | 61.75 | 58.16 | 54.16 | 51.10 | 48.53 | 46.54 | 44.20 | 42.28 | -2.57 |
| HardNet, $c = 50\%$ | 65.13 | 60.37 | 56.12 | 53.17 | 50.17 | 47.74 | 45.34 | 43.35 | 42.13 | -2.72 |
| HardNet, $c = 80\%$ | 69.73 | 64.46 | 60.42 | 57.09 | 54.09 | 51.18 | 48.76 | 46.81 | 45.66 | +0.81 |
| HardNet, $c = 90\%$ | 64.68 | 59.80 | 55.70 | 52.82 | 50.01 | 47.30 | 45.17 | 43.34 | 42.09 | -2.76 |
| SoftNet, $c = 50\%$ | 72.83 | 67.23 | 62.82 | 59.41 | 56.44 | 53.55 | 50.92 | 48.99 | 47.60 | +2.75 |
| SoftNet, $c = 80\%$ | 76.63 | 70.13 | 65.92 | 62.52 | **59.49** | **56.56** | 53.71 | 51.72 | **50.48** | +5.63 |
| SoftNet, $c = 90\%$ | 77.00 | **70.38** | 65.94 | 62.45 | 59.32 | 56.25 | **53.76** | **51.75** | 50.39 | +5.54 |
| SoftNet, $c = 97\%$ | **77.17** | 70.32 | **66.15** | **62.55** | 59.48 | 56.46 | 53.71 | 51.68 | 50.24 | +5.39 |

**Baselines.** We mainly compare our SoftNet with architecture-based methods for FSCIL: FSLL (Mazumder et al., 2021) that selects important parameters for each session, and HardNet, representing a binary subnetwork. Furthermore, we compare other FSCIL methods such as iCaRL (Rebuffi et al., 2017), Rebalance (Hou et al., 2019), TOPIC (Tao et al., 2020), IDLVQ-C (Chen and Lee, 2020), and F2M (Shi et al., 2021). We also include a joint training method (Shi et al., 2021) that uses all previously seen data, including the base and the following few-shot tasks for training as a reference. Furthermore, we fix the classifier re-training method (cRT) (Kang et al., 2019) for long-tailed classification trained with all encountered data as the approximated upper bound.

**Experimental details.** The experiments are conducted with NVIDIA GPU RTX8000 on CUDA 11.0. We also randomly split each dataset into multiple sessions. We run each algorithm ten times for each dataset and report their mean accuracy. We adopt ResNet18 (He et al., 2016) as the backbone network. For data augmentation, we use standard random crop and horizontal flips. In the base session training stage, we select top-$c\%$ weights at each layer and acquire the optimal soft-subnetworks with the best validation accuracy. In each incremental few-shot learning session, the total number of training epochs is 6, and the learning rate is 0.02. We train new class session samples using a few minor weights of the soft-subnetwork (Conv4x layer of ResNet18 and Conv3x layer of ResNet20) obtained by the base session learning. We specify further experiment details in Appendix A.

## 5.2 Results and Comparisons

We compared SoftNet with the architecture-based methods - FSLL and HardNet. We pick FSLL as an architecture-based baseline since it selects important parameters for acquiring old/new class knowledge. The architecture-based results on CIFAR-100 and miniImageNet are presented in Table 1 and Table 2 respectively. The performances of HardNet show the effectiveness of the subnetworks that go with less model capacity compared to dense networks. To emphasize our point, we found that ResNet18, with approximately 50% parameters, achieves comparable performances with FSLL on CIFAR-100 and miniImageNet. In addition, the performances of ResNet20 with 30% parameters (HardNet) are comparable with those of FSLL on CIFAR-100, as denoted in Appendix of Table 9 and Table 11, including performances (Figure 4 and Figure 5) and smoothness in t-SNE plots (Figure 6).

Experimental results are prepared to analyze the overall performances of SoftNet according to the sparsity and dataset as shown in Figure 2. As we increase the number of parameters employed by SoftNet, we achieve performance gain on both benchmark datasets. The performance variance of SoftNet's sparsity seems to be depending on datasets from the fact that the performance variance on CIFAR-100 is less than that on miniImageNet. In addition, SoftNet retains prior session knowledge successfully in both experiments as described in the dashed line, and the performances of SoftNet ($c = 60.0\%$) on the new class session (8, 9) of CIFAR-100 than those of SoftNet ($c = 80.0\%$) as depicted in the dashed-dot line. From these results, we could expect that the best performances

depend on the number of parameters and properties of datasets. We further result on comparisons of HardNet and SoftNet in Appendix B.

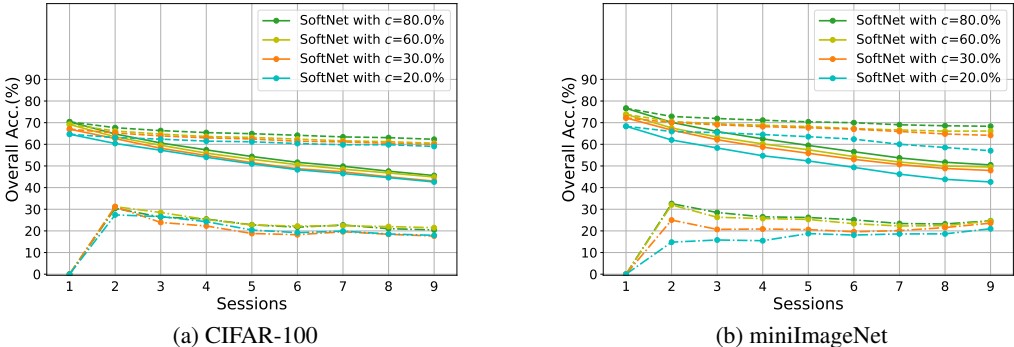

(a) CIFAR-100          (b) miniImageNet

Figure 2: **Classification accuracy of SoftNet on CIFAR-100 and miniImageNet for 5-way 5-shot FSCIL:** the overall performance depends on capacity $c$ and the softness of subnetwork. Note that solid(——), dashed(- - -), and dashed-dot(— · —) lines denote overall, base, and novel class performances respectively.

Our SoftNet outperforms the state-of-the-art methods and cRT, which is used as the approximate upper bound of FSCIL (Shi et al., 2021) as shown in Table 1 and Table 2. Moreover, Figure 3 represents the outstanding performances of SoftNet on CIFAR-100 and miniImageNet. SoftNet provides a new upper bound on each dataset, outperforming cRT, while HardNet provides new baselines among pruning-based methods.

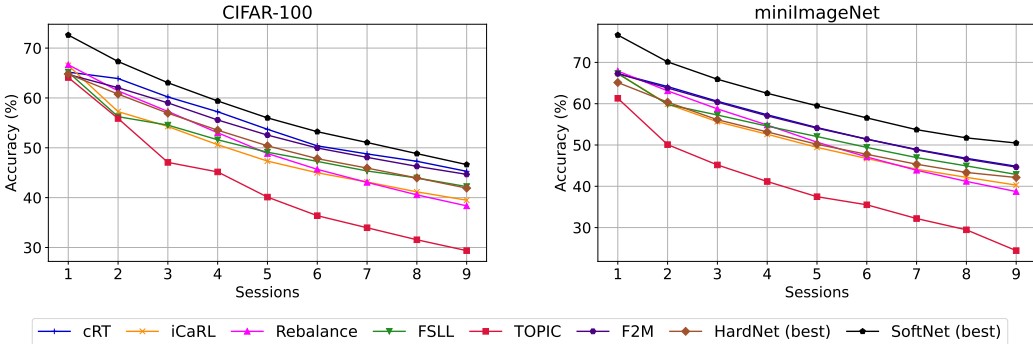

Figure 3: Comparision of subnetworks (HardNet and SoftNet) with state-of-the-art methods.

## 5.3 LAYER-WISE ACCURACY

In incremental few-shot learning sessions, we train new class session samples using a few minor weights $m_{\text{minor}}$ of the specific layer. At the same, we entirely fix the remaining weights to investigate the best performances as shown in Table 3. The best performances involve fine-tuning at the Conv5x layer with $c = 97\%$. It means features computed by the lower layer are general and reusable in different classes. On the other hand, features from the higher layer are specific and highly dependent on the dataset.

## 5.4 ARCHITECTURE-WISE ACCURACY

Depending on architectures, the performances of subnetworks vary, and the sparsity is also one another: ResNet18 tends to use dense parameters, whereas ResNet20 tends to use sparse parameters on CIFAR-100 for 5-way 5-shot as shown in Table 4. We observed that the SoftNet with ResNet20 has a more sparse solution as $c = 90\%$ than ResNet18 on this CIFAR-100 FSCIL setting. From these

Table 3: Classification accuracy of ResNet18 on miniImageNet for 5-way 5-shot incremental learning. The layer-wise inspection with fixed $c = 97\%$. *all* denotes that all minor weights $m_{minor}$ of the entire layers were trained while only *conv∗x* trained.

| Method | sessions | | | | | | | | | The gap with cRT |
|---|---|---|---|---|---|---|---|---|---|---|
| | 1 | 2 | 3 | 4 | 5 | 6 | 7 | 8 | 9 | |
| cRT (Shi et al., 2021) | 67.30 | 64.15 | 60.59 | 57.32 | 54.22 | 51.43 | 48.92 | 46.78 | 44.85 | - |
| SoftNet, Conv2x | 77.17 | 70.29 | 66.09 | 62.54 | 59.44 | 56.43 | 53.68 | 51.60 | 50.19 | +5.34 |
| SoftNet, Conv3x | 77.17 | 70.30 | 66.05 | 62.51 | 59.42 | 56.44 | 53.70 | 51.59 | 50.15 | +5.30 |
| SoftNet, Conv4x | 77.17 | 70.30 | 66.08 | 62.54 | 59.45 | 56.46 | 53.70 | 51.59 | 50.17 | +5.32 |
| SoftNet, Conv5x | **77.17** | **70.32** | **66.15** | **62.55** | **59.48** | **56.46** | **53.71** | **51.68** | **50.24** | **+5.39** |
| SoftNet, All | 77.17 | 65.09 | 55.25 | 45.92 | 38.20 | 33.37 | 29.52 | 27.66 | 25.64 | -19.21 |

observations, our SoftNet could significantly impact deep neural network architecture search - it helps to search sparse and task-specific architecture.

Table 4: Classification accuracy of ResNet18, 20, 32, and 50 on CIFAR-100 for 5-way 5-shot FSCIL.

| Method | sessions | | | | | | | | | The gap with cRT |
|---|---|---|---|---|---|---|---|---|---|---|
| | 1 | 2 | 3 | 4 | 5 | 6 | 7 | 8 | 9 | |
| ResNet18, cRT (Shi et al., 2021) | 65.18 | 63.89 | 60.20 | 57.23 | 53.71 | 50.39 | 48.77 | 47.29 | 45.28 | - |
| ResNet18, SoftNet, $c = 70\%$ | 70.92 | 65.16 | 61.00 | 57.25 | 54.09 | 51.37 | 49.29 | 47.03 | 44.90 | -0.38 |
| SoftNet, $c = 90\%$ | 72.25 | 66.82 | 62.63 | 58.98 | 55.64 | 52.77 | 50.71 | 48.42 | 46.15 | +0.87 |
| SoftNet, $c = 99\%$ | 72.62 | 67.31 | 63.05 | 59.39 | 56.00 | 53.23 | 51.06 | 48.83 | 46.63 | +1.35 |
| ResNet20, SoftNet, $c = 70\%$ | 70.38 | 66.16 | 62.63 | 58.93 | 55.81 | 53.11 | 51.38 | 49.29 | 47.08 | +1.80 |
| SoftNet, $c = 90\%$ | 72.63 | 68.60 | 64.96 | 61.25 | 57.98 | 55.32 | 53.48 | 51.46 | 49.20 | +3.92 |
| SoftNet, $c = 99\%$ | 71.78 | 67.79 | 63.86 | 60.07 | 57.05 | 54.32 | 52.34 | 50.28 | 48.11 | +2.83 |
| ResNet32, SoftNet, $c = 90\%$ | 75.47 | 70.84 | 66.84 | 63.01 | 59.69 | 56.86 | 54.75 | 52.70 | 50.36 | +5.08 |
| SoftNet, $c = 93\%$ | 75.35 | 71.22 | 67.25 | 63.25 | 60.05 | 57.24 | 55.16 | 53.01 | 50.76 | +5.48 |
| SoftNet, $c = 95\%$ | 74.67 | 70.32 | 66.31 | 62.61 | 59.29 | 56.54 | 54.52 | 52.30 | 50.05 | +4.77 |
| ResNet50, SoftNet, $c = 70\%$ | 76.20 | 71.82 | 67.90 | 64.17 | 60.91 | 57.89 | 55.72 | 53.20 | 50.87 | +5.59 |
| SoftNet, $c = 80\%$ | **78.20** | **73.32** | **69.22** | **65.43** | **62.09** | **59.08** | **56.80** | **54.45** | **52.18** | **+6.90** |
| SoftNet, $c = 90\%$ | 77.67 | 72.73 | 68.50 | 64.57 | 61.08 | 58.06 | 55.70 | 53.37 | 51.20 | +5.92 |

## 5.5 DISCUSSIONS

Based on our thorough empirical study, we uncover the following facts: (1) Depending on architectures, the performances of subnetworks vary, and the sparsity is also one another: ResNet18 tends to use dense parameters, while ResNet20 tends to use sparse parameters on CIFAR-100 FSCIL settings. This result provides the general pruning-based model with a hidden clue. (2) In general, fine-tuning strategies are essential in retaining prior knowledge and learning new knowledge. We found that performance varies depending on fine-tuning a Conv layer through the layer-wise inspection. Lastly, (3) from overall experimental results, the base session learning is significant for lifelong learners to acquire generalized performances in FSCIL.

## 6 CONCLUSION

Inspired by *Regularized Lottery Ticket Hypothesis (RLTH)*, which hypothesizes that smooth subnetworks exist within a dense network, we propose *Soft-SubNetworks (SoftNet)*; an incremental learning strategy that preserves the learned class knowledge and learns the newer ones. More specifically, *SoftNet* jointly learned the model weights and adaptive soft masks to minimize catastrophic forgetting and to avoid overfitting novel few samples in FSCIL. Finally, we compared a comprehensive empirical study on *SoftNet* with multiple class incremental learning methods. Extensive experiments on benchmark tasks demonstrate how our method achieves superior performance over the state-of-the-art class incremental learning methodologies. We also discovered how subnetworks perform differently under specified architectures and datasets through ablation studies. In addition, we emphasized the importance of fine-tuning and base session learning in achieving optimum performance for FSCIL. We believe that our findings could bring a monumental on deep neural network architecture search, both on task-specific architectures and utilization of sparse models.

**Acknowledgement.**This work was supported by Institute for Information & communications Technology Promotion (IITP) grant funded by the Korea government (MSIT) (No. 2021-0-01381, Development of Causal AI through Video Understanding and Reinforcement Learning, and Its Applications to Real Environments) and partly supported by Institute of Information & communications Technology Planning & Evaluation (IITP) grant funded by the Korea government (MSIT) (No. 2022-0-00184, Development and Study of AI Technologies to Inexpensively Conform to Evolving Policy on Ethics).

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

# A    EXPERIMENTAL DETAILS

We validate the effectiveness of the soft-subnetwork in our method on several benchmark datasets against various architecture-based methods for Few-Shot Class Incremental Learning (FSCIL). To proceed with the details of our experiments, we first explain the datasets and how we involve them in our experiments. Later, we detail experiment setups, including architecture details, preprocessing, and training budget.

## A.1    DATASETS

The following datasets are utilized for comparisons:

**CIFAR-100** In CIFAR-100, each class contains $500$ images for training and $100$ images for testing. Each image has a size of $32 \times 32$. Here, we follow an identical FSCIL procedure as in (Shi et al., 2021), where we divide the dataset into a base session with 60 base classes and eight novel sessions with a 5-way 5-shot problem on each session.

**miniImageNet** miniImageNet consists of RGB images from 100 different classes, where each class contains $500$ training images and $100$ test images of size $84 \times 84$. Originally proposed for few-shot learning problems, miniImageNet is part of a much larger ImageNet dataset. Compared with CIFAR-100, the miniImageNet dataset is more complex and suitable for prototyping. The setup of miniImageNet is similar to that of CIFAR-100. To proceed with our evaluation, we follow the procedure described in (Shi et al., 2021), where we incorporate 60 base classes and eight novel sessions through 5-way 5-shot problems.

**CUB-200-2011** CUB-200-2011 contains $200$ fine-grained bird species with $11,788$ images with varying images for each class. To proceed with experiments, we split the dataset into $6,000$ training images and $6,000$ test images as in (Tao et al., 2020). During training, We randomly crop each image to be of size $224 \times 224$. We fix the first 100 classes as base classes, where we utilize all samples in these respective classes to train the model. On the other hand, we treat the remaining 100 classes as novel categories split into ten novel sessions with a 10-way 5-shot problem in each session.

## A.2    EXPERIMENT SETUPS

We begin this section by describing the setups used for experiments in CIFAR-100 and miniImageNet. After that, we proceed with a follow-up discussion on the configuration we employ for experiments involving the CUB-200-2011 dataset.

**CIFAR-100 and miniImageNet.** For experiments in these two datasets, we are using NVIDIA GPU RTX8000 on CUDA 11.0. We randomly split these two datasets into multiple sessions, as described in the previous sub-section. We run each algorithm ten times for experiments on both datasets with a fixed split and report their mean accuracy. We adopt ResNet18 (He et al., 2016) as the backbone network. For data augmentation, we use standard random crop and horizontal flips. During the training stage in the base session, we select top-$c\%$ weights at each layer and acquire the optimal soft-subnetworks with the best validation accuracy. For each incremental few-shot learning session, we train our model for six epochs with a learning rate is $0.02$. We train new class session samples using a few minor weights of the soft-subnetwork (conv4x layer of ResNet18 and conv3x layer of ResNet20) obtained by learning at the base session.

**CUB-200-2011.** Besides experiments in the previous two datasets, we conducted an additional experiment on this dataset. We prepare this dataset following the split procedure described in the previous sub-section. We run each algorithm ten times and report their mean accuracy. We also adopt ResNet18 (He et al., 2016) as the backbone network and follow the same data augmentation as in the previous two datasets. We follow the same base-session training procedure as in the other two datasets. In each incremental few-shot learning session $t > 1$, the total number of training epochs is $10$, and the learning rate is $0.1$. We train new class session samples using a few minor weights of the soft-subnetwork (conv4x layer of ResNet18) obtained at the base session.

## B    RESULTS AND CONCLUSIONS

To expand upon the results of our paper, we conduct more experiments on various datasets mentioned in the previous section. We first display the full performance table with more capacity values $c$ employed towards our method in Table 9 and Table 11. Next, we identify how choosing a different architecture would impact the performance of our algorithm in Table 10. Furthermore, we analyze the performance of our method on the CUB-200-2011 dataset in Table 7.

Through extensive experiments, we deduce the following three conclusions for incorporating our method in the few-shot class incremental learning:

**Structure.** We identified a SubNetwork of ResNet18 and ResNet20 with varying capacities on CIFAR-100 for the 5-way 5-shot FSCIL setting as shown in Table 9 and Table 10. First, according to both tables, our method performs better as we use more parameters within our network. In addition, as denoted in our paper, we see how effective subnetwork is by observing how HardNet, with only 50% of its dense capacity, achieves comparable performance to methods utilizing dense networks, while SoftNet can do the same with only 30% of its dense capacity. Furthermore, we argue that our method is architecture-dependent. Our observation from Table 10 shows that at ResNet18, our architecture performs the best at the maximum capacity of $c = 99\%$, while at ResNet20, we achieve the optimum performance at $c = 90\%$.

**Comparisions of Hard and SoftNet**. Furthermore, increasing the number of network parameters leads to better overall performance in both subnetworks types, as shown in Figure 4 and Figure 5. Subnetworks, in the form of HardNet and SoftNet, tend to retain prior (base) session knowledge denoted in dashed (- - - -) line, and HardNet seems to be able to classify new session class samples without continuous updates stated in dashed-dot ($—\cdot—$) line. From this, we could expect how much previous knowledge HardNet learned at the base session to help learn new incoming tasks (Forward Transfer). The overall performances of SoftNet are better than HardNet since SoftNet improves both base/new session knowledge by updating minor subnetworks. Subnetworks have a broader spectrum of performances on miniImageNet (Figure 5) than on CIFAR-100 (Figure 4). This could be an observation caused by the dataset complexity - i.e., if the miniImagenet dataset is more complex or harder to learn for a subnetwork or a deep model as such subnetworks need more parameters to learn miniImageNet than the CIFAR-100 dataset.

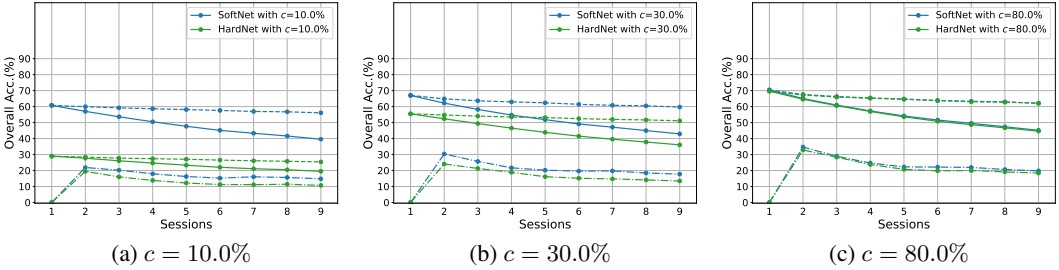

(a) $c = 10.0\%$          (b) $c = 30.0\%$          (c) $c = 80.0\%$

Figure 4: **Performances of HardNet v.s. SoftNet on CIFAR-100 for 5-way 5-shot FSCIL:** the overall performance depends on capacity $c$ and the softness of subnetwork. Note that solid(——), dashed(- - - ), and dashed-dot($—\cdot—$) lines denote overall, base, and novel class performances respectively.

**Smoothness of SoftNet.** As emphasized in Table 11, SoftNet has a broader spectrum of performances than HardNet on miniImageNet. 20% of minor subnet might provide a smoother representation than HardNet because the performance of SoftNet was the best approximately at $c = 80\%$. From these results, we could expect that model parameter smoothness guarantees quite competitive results. To support the claim, we prepared the loss landscapes of a dense neural network, HardNet, and SoftNet on two Hessian eigenvectors (Yao et al., 2020) as shown in Fig. 7. We observed the following points through simple experiments:

From these results, we can expect how much knowledge the specified subnetworks can retain and acquire on each dataset.

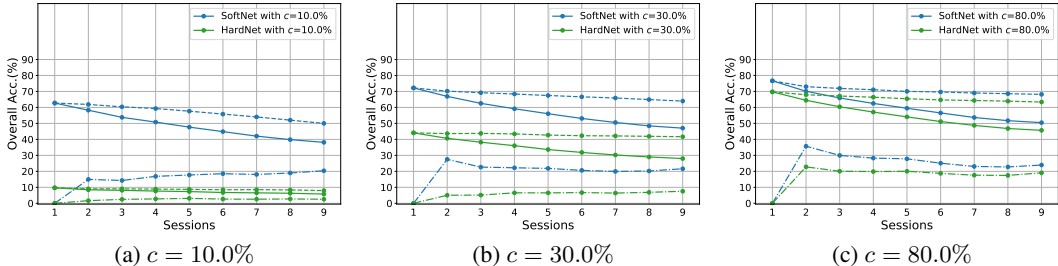

(a) $c = 10.0\%$          (b) $c = 30.0\%$          (c) $c = 80.0\%$

Figure 5: **Performances of HardNet v.s. SoftNet on miniImageNet for 5-way 5-shot FSCIL:** the overall performance depends on capacity $c$ and the softness of subnetwork. Note that solid(——), dashed(- - -), and dashed-dot(— · —) lines denote overall, base, and novel class performances respectively.

- The loss landscapes of Subnetworks (HardNet and SoftNet) were flatter than those of dense neural networks.

- The minor subnet of SoftNet helped find a flat global minimum despite random scaling weights in the training process.

Moreover, we compared the embeddings using t-SNE plots as shown in Figure 6. In t-SNE's 2D embedding spaces, the overall discriminative of SoftNet is better than that of HardNet in terms of base class set and novel class set. This 70% of minor subnet affects SoftNet positively in base session training and offers good initialized weights in novel session training.

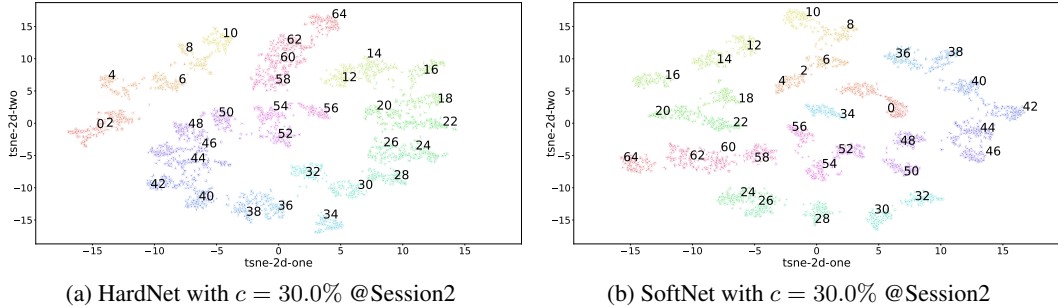

(a) HardNet with $c = 30.0\%$ @Session2       (b) SoftNet with $c = 30.0\%$ @Session2

Figure 6: **t-SNE Plots of HardNet v.s. SoftNet on miniImageNet for 5-way 5-shot FSCIL:** t-SNE plots represent the embeddings of the even-numbered test class samples and compare one another. Note Session1 Class Set:$\{0, \cdots, 59\}$ and Session2 Novel Class Set:$\{60, \cdots, 64\}$.

**Preciseness.** Regarding fine-grained and small-sized CUB200-2011 FSCIL settings, HardNet also shows comparable results with the baselines, and SoftNet outperforms others as denoted in Table 7. In this FSCIL setting, we acquired the best performances of SoftNet through the specific parameter selections. As of now, our SoftNet achieves state-of-the-art results on the three datasets.

## C    CONVERGENCE OF SUBNETWORKS

**Convergences of HardNet and SoftNet.** To interpret the convergence of SoftNet, we follow the Lipschitz-continuous objective gradients (Bottou et al., 2018): the objective function of dense networks $R : \mathbb{R}^d \to \mathbb{R}$ is continuously differentiable and the gradient function of $R$, namely, $\nabla R : \mathbb{R}^d \to \mathbb{R}^d$, *Lipschitz continuous with Lipschitz constant $L > 0$*, i.e.,

$$||\nabla R(\boldsymbol{\theta}) - \nabla R(\boldsymbol{\theta}')||_2 \leq L||\boldsymbol{\theta} - \boldsymbol{\theta}'|| \quad \text{for all } \{\boldsymbol{\theta}, \boldsymbol{\theta}'\} \subset \mathbb{R}^d. \tag{6}$$

Following the same formula, we define the Lipschitz-continuous objective gradients of subnetworks as follows:

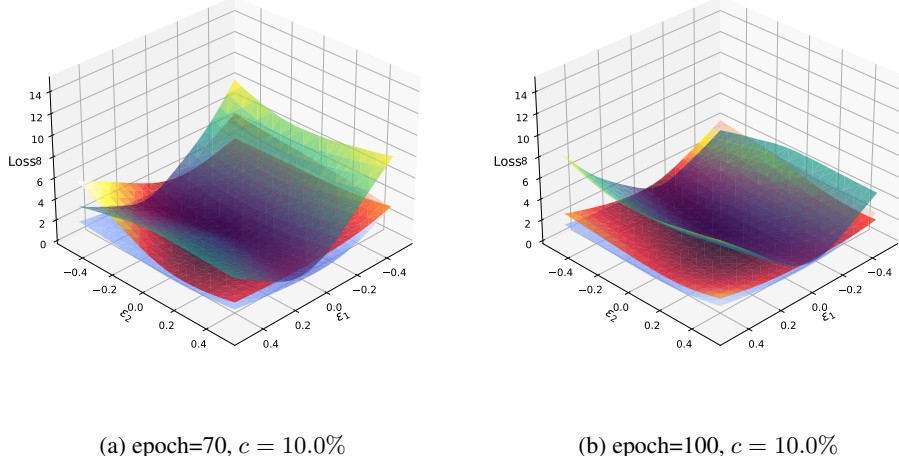

(a) epoch=70, $c = 10.0\%$           (b) epoch=100, $c = 10.0\%$

Figure 7: **Loss landscapes of** DenseNet, HardNet, **and** SoftNet: Subnetworks provide a more flat global minimum than dense neural networks. To demonstrate the loss landscapes, we trained a simple three-layered, fully connected model (fc-4-25-30-3) on the Iris Flower dataset (which is three classification problem) for 100 epochs.

$$||\nabla R(\boldsymbol{\theta} \odot \boldsymbol{m}) - \nabla R(\boldsymbol{\theta}' \odot \boldsymbol{m})||_2 \leq L||(\boldsymbol{\theta} - \boldsymbol{\theta}') \odot \boldsymbol{m}|| \quad \text{for all } \{\boldsymbol{\theta}, \boldsymbol{\theta}'\} \subset \mathbb{R}^d. \tag{7}$$

where $\boldsymbol{m}$ is a binary mask. In comparision of Eq. 6 and 7, we use the theoretical analysis (Ye et al., 2020) where subnetwork achieve a faster rate of $R(\boldsymbol{\theta} \odot \boldsymbol{m}) = \mathcal{O}(1/||\boldsymbol{m}||_1^2)$ at most. The comparison is as follows:

$$\frac{||\nabla R(\boldsymbol{\theta} \odot \boldsymbol{m}) - \nabla R(\boldsymbol{\theta}' \odot \boldsymbol{m})||_2}{||(\boldsymbol{\theta} - \boldsymbol{\theta}') \odot \boldsymbol{m}||} < \frac{||\nabla R(\boldsymbol{\theta}) - \nabla R(\boldsymbol{\theta}')||_2}{||\boldsymbol{\theta} - \boldsymbol{\theta}'||} \leq L \tag{8}$$

The smaller the value is, the flatter the solution (loss landscape) has. The equation is established from the relationship $R(\boldsymbol{\theta} \odot \boldsymbol{m}) \ll R^*(\boldsymbol{\theta})$, where $R^*(\boldsymbol{\theta}$ denotes the best possible loss achievable by convex combinations of all parameters despite $||(\boldsymbol{\theta} - \boldsymbol{\theta}') \odot \boldsymbol{m}|| < ||\boldsymbol{\theta} - \boldsymbol{\theta}'||$. Furthermore, we have the following inequality if $||R(\boldsymbol{\theta} \odot \boldsymbol{m}_{hard}) - R(\boldsymbol{\theta} \odot \boldsymbol{m}_{soft})|| \simeq 0$ and $||\boldsymbol{m}_{hard}|| < ||\boldsymbol{m}_{soft}||$:

$$\frac{||\nabla R(\boldsymbol{\theta} \odot \boldsymbol{m}_{hard}) - \nabla R(\boldsymbol{\theta}' \odot \boldsymbol{m}_{hard})||_2}{||(\boldsymbol{\theta} - \boldsymbol{\theta}') \odot \boldsymbol{m}_{hard}||} \geq \frac{||\nabla R(\boldsymbol{\theta} \odot \boldsymbol{m}_{soft}) - \nabla R(\boldsymbol{\theta}' \odot \boldsymbol{m}_{soft})||_2}{||(\boldsymbol{\theta} - \boldsymbol{\theta}') \odot \boldsymbol{m}_{soft}||} \tag{9}$$

where the equality holds iff $||\boldsymbol{m}_{hard}|| = ||\boldsymbol{m}_{soft}||$. We prepare the loss landscapes of Dense Network, Hard-WSN, and Soft-WSN as shown in Figure 7 as an example to support the inequality.

## D ADDITIONAL COMPARISONS WITH CURRENT WORKS

**Comparisons with SOTA.** We compare SoftNet with the following state-of-art-methods on TOPIC class split (Tao et al., 2020) of three benchmark datasets - CIFAR100 (Table 5), miniImageNet (Table 6), and CUB-200-2011 (Table 7). We summarize the current FSCIL methods as follows:

- **CEC** Zhang et al. (2021): The authors proposed a Continually Evolved Classifier (CEC) that employs a graph model to propagate context information between classifiers for adaptation.

- **LIMIT** Zhou et al. (2022): The authors proposed a new paradigm for FSCIL based on meta-learning by LearnIng Multi-phase Incremental Tasks (LIMIT), which synthesizes fake FSCIL tasks from the base dataset. Besides, LIMIT also constructs a calibration module based on a transformer, which calibrates the old class classifiers and new class prototypes into the same scale and fills in the semantic gap.

- **MetaFSCIL** Chi et al. (2022): The authors proposed a bilevel optimization based on meta-learning to directly optimize the network to learn how to learn incrementally in the setting of FSCIL. Concretely, They proposed to sample sequences of incremental tasks from base classes for training to simulate the evaluation protocol. For each task, the model is learned using a meta-objective to perform fast adaptation without forgetting. Furthermore, they proposed a bi-directional guided modulation to modulate activations and reduce catastrophic forgetting.

- **C-FSCIL** Hersche et al. (2022): The authors proposed C-FSCIL, which is architecturally composed of a frozen meta-learned feature extractor, a trainable fixed-size fully connected layer, and a rewritable dynamically growing memory that stores as many vectors as the number of encountered classes.

- **Subspace Reg.** Akyürek et al. (2021): The authors presented a straightforward approach that enables using logistic regression classifiers for few-shot incremental learning. The key to this approach is a new family of subspace regularization schemes that encourage weight vectors for new classes to lie close to the subspace spanned by the weights of existing classes.

- **Entropy-Reg** Liu et al. (2022): The authors alternatively proposed using data-free replay to synthesize data by a generator without accessing real data.

- **ALICE** Peng et al. (2022): The authors proposed a method - Augmented Angular Loss Incremental Classification or ALICE - inspired by the similarity of the goals for FSCIL and modern face recognition systems. Instead of the commonly used cross-entropy loss, they proposed using the angular penalty loss to obtain well-clustered features in ALICE.

Table 5: Classification accuracy of ResNet18 on CIFAR-100 for 5-way 5-shot incremental learning with the same class split as in TOPIC (Cheraghian et al., 2021). $*$ denotes the results reported from Shi et al. (2021). $\dagger$ represents our reproduced results.

| Method | sessions | | | | | | | | | The gap with cRT |
|---|---|---|---|---|---|---|---|---|---|---|
| | 1 | 2 | 3 | 4 | 5 | 6 | 7 | 8 | 9 | |
| cRT Shi et al. (2021)* | 72.28 | 69.58 | 65.16 | 61.41 | 58.83 | 55.87 | 53.28 | 51.38 | 49.51 | |
| Joint-training Shi et al. (2021)* | 72.28 | 68.40 | 63.31 | 59.16 | 55.73 | 52.81 | 49.01 | 46.74 | 44.34 | -5.17 |
| Baseline Shi et al. (2021) | 72.28 | 68.01 | 64.18 | 60.56 | 57.44 | 54.69 | 52.98 | 50.80 | 48.70 | -0.81 |
| iCaRL Rebuffi et al. (2017)* | 72.05 | 65.35 | 61.55 | 57.83 | 54.61 | 51.74 | 49.71 | 47.49 | 45.03 | -4.48 |
| Rebalance Hou et al. (2019)* | 74.45 | 67.74 | 62.72 | 57.14 | 52.78 | 48.62 | 45.56 | 42.43 | 39.22 | -10.29 |
| FSLL Mazumder et al. (2021)* | 72.28 | 63.84 | 59.64 | 55.49 | 53.21 | 51.77 | 50.93 | 48.94 | 46.96 | -2.55 |
| iCaRL Rebuffi et al. (2017) | 64.10 | 53.28 | 41.69 | 34.13 | 27.93 | 25.06 | 20.41 | 15.48 | 13.73 | -35.78 |
| Rebalance Hou et al. (2019) | 64.10 | 53.05 | 43.96 | 36.97 | 31.61 | 26.73 | 21.23 | 16.78 | 13.54 | -35.97 |
| TOPIC Cheraghian et al. (2021) | 64.10 | 55.88 | 47.07 | 45.16 | 40.11 | 36.38 | 33.96 | 31.55 | 29.37 | -20.14 |
| CEC Zhang et al. (2021) | 73.07 | 68.88 | 65.26 | 61.19 | 58.09 | 55.57 | 53.22 | 51.34 | 49.14 | -0.37 |
| F2M Shi et al. (2021) | 71.45 | 68.10 | 64.43 | 60.80 | 57.76 | 55.26 | 53.53 | 51.57 | 49.35 | -0.16 |
| LIMIT Zhou et al. (2022) | 73.81 | 72.09 | 67.87 | 63.89 | 60.70 | 57.77 | 55.67 | 53.52 | 51.23 | +1.72 |
| MetaFSCIL Chi et al. (2022) | 74.50 | 70.10 | 66.84 | 62.77 | 59.48 | 56.52 | 54.36 | 52.56 | 49.97 | +0.46 |
| ALICE Peng et al. (2022) | 79.00 | 70.50 | 67.10 | 63.40 | 61.20 | 59.20 | 58.10 | 56.30 | 54.10 | +4.59 |
| Entropy-Reg Liu et al. (2022) | 74.40 | 70.20 | 66.54 | 62.51 | 59.71 | 56.58 | 54.52 | 52.39 | 50.14 | +0.63 |
| C-FSCIL Hersche et al. (2022) | 77.50 | 72.45 | 67.94 | 63.80 | 60.24 | 57.34 | 54.61 | 52.41 | 50.23 | +0.72 |
| FSLL Mazumder et al. (2021) | 64.10 | 55.85 | 51.71 | 48.59 | 45.34 | 43.25 | 41.52 | 39.81 | 38.16 | -11.35 |
| FSLL+SS Mazumder et al. (2021) | 66.76 | 55.52 | 52.20 | 49.17 | 46.23 | 44.64 | 43.07 | 41.20 | 39.57 | -9.94 |
| HardNet, $c = 50\%$ | 78.35 | 74.12 | 70.13 | 65.88 | 62.74 | 59.56 | 57.98 | 56.31 | 54.32 | +4.81 |
| HardNet, $c = 80\%$ | 79.27 | 75.38 | 71.11 | 66.68 | 63.32 | 60.06 | 58.16 | 56.40 | 54.31 | +4.80 |
| HardNet, $c = 90\%$ | 79.22 | 74.77 | 70.89 | 66.41 | 62.90 | 59.48 | 58.10 | 56.13 | 53.92 | +4.41 |
| SoftNet, $c = 50\%$ | 79.88 | 75.54 | 71.64 | 67.47 | 64.45 | 61.09 | 59.07 | 57.29 | **55.33** | **+5.82** |
| SoftNet, $c = 80\%$ | **80.33** | **76.23** | **72.19** | **67.83** | **64.64** | **61.39** | **59.32** | **57.37** | 54.94 | +5.43 |
| SoftNet, $c = 90\%$ | 79.97 | 75.75 | 71.76 | 67.36 | 64.09 | 60.91 | 59.07 | 56.94 | 54.76 | +5.25 |

Leveraged by regularized backbone ResNet, SoftNet outperformed all existing current works on CIFAR100 as shown in Table 5. On miniImageNet Table 6 and CUB-200-201 Table 7, the performances of SoftNet were comparable with those of ALICE and LIMIT, considering that ALICE used class/data augmentations and LIMIT added an extra multi-head attention layer.

**Comparisions of SoftNet and AANet**. Our SoftNet and AANet Liu et al. (2021) have proposed alleviating catastrophic forgetting in FSCIL and CIL, respectively. AANet consists of multi-ResNets: one residual block learns new knowledge while another fine-tunes to maintain the previously learned knowledge. Through the learnable scaling parameter for the linear combination of the multi-ResNet features, AANet showed outstanding performances in the CSIL setting. However, AANet tends to overfit since the ResNet block's parameters are fully used to update a few new class data in FSCIL.

Table 6: Classification accuracy of ResNet18 on miniImageNet for 5-way 5-shot incremental learning with the same class split as in TOPIC (Cheraghian et al., 2021). * denotes results reported from Shi et al. (2021). † represents our reproduced results.

| Method | sessions | | | | | | | | | The gap with cRT |
|---|---|---|---|---|---|---|---|---|---|---|
| | 1 | 2 | 3 | 4 | 5 | 6 | 7 | 8 | 9 | |
| cRT Shi et al. (2021)* | 72.08 | 68.15 | 63.06 | 61.12 | 56.57 | 54.47 | 51.81 | 49.86 | 48.31 | - |
| Joint-training Shi et al. (2021)* | 72.08 | 67.31 | 62.04 | 58.51 | 54.41 | 51.53 | 48.70 | 45.49 | 43.88 | -4.43 |
| Baseline Shi et al. (2021) | 72.08 | 66.29 | 61.99 | 58.71 | 55.73 | 53.04 | 50.40 | 48.59 | 47.31 | -1.0 |
| iCaRL Rebuffi et al. (2017)* | 71.77 | 61.85 | 58.12 | 54.60 | 51.49 | 48.47 | 45.90 | 44.19 | 42.71 | -5.6 |
| Rebalance Hou et al. (2019)* | 72.30 | 66.37 | 61.00 | 56.93 | 53.31 | 49.93 | 46.47 | 44.13 | 42.19 | -6.12 |
| FSLL Mazumder et al. (2021)* | 72.08 | 59.04 | 53.75 | 51.17 | 49.11 | 47.21 | 45.35 | 44.06 | 43.65 | -4.66 |
| iCaRL Rebuffi et al. (2017) | 61.31 | 46.32 | 42.94 | 37.63 | 30.49 | 24.00 | 20.89 | 18.80 | 17.21 | -31.10 |
| Rebalance Hou et al. (2019) | 61.31 | 47.80 | 39.31 | 31.91 | 25.68 | 21.35 | 18.67 | 17.24 | 14.17 | -34.14 |
| TOPIC Cheraghian et al. (2021) | 61.31 | 50.09 | 45.17 | 41.16 | 37.48 | 35.52 | 32.19 | 29.46 | 24.42 | -23.89 |
| IDLVQ-C Chen and Lee (2020) | 64.77 | 59.87 | 55.93 | 52.62 | 49.88 | 47.55 | 44.83 | 43.14 | 41.84 | -6.47 |
| CEC Zhang et al. (2021) | 72.00 | 66.83 | 62.97 | 59.43 | 56.70 | 53.73 | 51.19 | 49.24 | 47.63 | -0.68 |
| F2M Shi et al. (2021) | 72.05 | 67.47 | 63.16 | 59.70 | 56.71 | 53.77 | 51.11 | 49.21 | 47.84 | -0.43 |
| LIMIT Zhou et al. (2022) | 73.81 | 72.09 | 67.87 | 63.89 | 60.70 | 57.77 | 55.67 | 53.52 | 51.23 | +2.92 |
| MetaFSCIL Chi et al. (2022) | 72.04 | 67.94 | 63.77 | 60.29 | 57.58 | 55.16 | 52.90 | 50.79 | 49.19 | +0.88 |
| ALICE Peng et al. (2022) | **80.60** | 70.60 | 67.40 | 64.50 | 62.50 | 60.00 | 57.80 | **56.80** | **55.70** | **+7.39** |
| C-FSCIL Hersche et al. (2022) | 76.40 | 71.14 | 66.46 | 63.29 | 60.42 | 57.46 | 54.78 | 53.11 | 51.41 | +3.10 |
| Entropy-Reg Liu et al. (2022) | 71.84 | 67.12 | 63.21 | 59.77 | 57.01 | 53.95 | 51.55 | 49.52 | 48.21 | -0.10 |
| Subspace Reg. Akyürek et al. (2021) | 80.37 | 71.69 | 66.94 | 62.53 | 58.90 | 55.00 | 51.94 | 49.76 | 46.79 | -1.52 |
| FSLL Mazumder et al. (2021) | 66.48 | 61.75 | 58.16 | 54.16 | 51.10 | 48.53 | 46.54 | 44.20 | 42.28 | -6.03 |
| FSLL+SS Mazumder et al. (2021) | 68.85 | 63.14 | 59.24 | 55.23 | 52.24 | 49.65 | 47.74 | 45.23 | 43.92 | -4.39 |
| HardNet, $c = 80\%$ | 78.70 | 72.55 | 68.26 | 64.45 | 61.74 | 58.93 | 55.99 | 54.09 | 52.74 | +4.43 |
| HardNet, $c = 87\%$ | 79.17 | 73.05 | 69.16 | 65.43 | 62.61 | 59.31 | 56.73 | 54.69 | 53.47 | +5.16 |
| HardNet, $c = 90\%$ | 79.15 | 72.03 | 68.76 | 65.32 | 62.00 | 58.21 | 56.52 | 53.66 | 53.07 | +4.76 |
| SoftNet, $c = 80\%$ | 79.37 | 74.31 | 69.89 | 66.16 | 63.40 | 60.75 | 57.62 | 55.67 | 54.34 | +6.03 |
| SoftNet, $c = 87\%$ | 79.77 | **75.08** | **70.59** | **66.93** | **64.00** | **61.00** | **57.81** | 55.81 | 54.68 | +6.37 |
| SoftNet, $c = 90\%$ | 79.72 | 74.25 | 70.00 | 66.35 | 63.19 | 60.04 | 57.36 | 55.38 | 54.14 | +5.83 |

Table 7: Classification accuracy of ResNet18 on CUB-200-2011 for 10-way 5-shot incremental learning (TOPIC class split Tao et al. (2020)). * denotes results reported from Shi et al. (2021). † represents our reproduced results.

| Method | sessions | | | | | | | | | | | The gap with cRT |
|---|---|---|---|---|---|---|---|---|---|---|---|---|
| | 1 | 2 | 3 | 4 | 5 | 6 | 7 | 8 | 9 | 10 | 11 | |
| cRT Shi et al. (2021)* | 77.16 | 74.41 | 71.31 | 68.08 | 65.57 | 63.08 | 62.44 | 61.29 | 60.12 | 59.85 | 59.30 | - |
| Joint-training Shi et al. (2021) | 77.16 | 74.39 | 69.83 | 67.17 | 64.72 | 62.25 | 59.77 | 59.05 | 57.99 | 57.81 | 56.82 | -2.48 |
| Baseline Shi et al. (2021) | 77.16 | 74.00 | 70.21 | 66.07 | 63.90 | 61.35 | 60.01 | 58.66 | 56.33 | 56.12 | 55.07 | -4.23 |
| iCaRL Rebuffi et al. (2017)* | 75.95 | 60.90 | 57.65 | 54.51 | 50.83 | 48.21 | 46.95 | 45.74 | 43.21 | 43.01 | 41.27 | -18.03 |
| Rebalance Hou et al. (2019)* | 77.44 | 58.10 | 50.15 | 44.80 | 39.12 | 34.44 | 31.73 | 29.75 | 27.56 | 26.93 | 25.30 | -34.00 |
| FSLL Mazumder et al. (2021)* | 77.16 | 71.85 | 66.53 | 59.95 | 58.01 | 57.00 | 56.06 | 54.78 | 52.24 | 52.01 | 51.47 | -7.83 |
| iCaRL Rebuffi et al. (2017) | 68.68 | 52.65 | 48.61 | 44.16 | 36.62 | 29.52 | 27.83 | 26.26 | 24.01 | 23.89 | 21.16 | -39.92 |
| Rebalance Hou et al. (2019) | 68.68 | 57.12 | 44.21 | 28.78 | 26.71 | 25.66 | 24.62 | 21.52 | 20.12 | 20.06 | 19.87 | -41.21 |
| TOPIC Cheraghian et al. (2021) | 68.68 | 62.49 | 54.81 | 49.99 | 45.25 | 41.40 | 38.35 | 35.36 | 32.22 | 28.31 | 26.28 | -34.80 |
| SPPR Zhu et al. (2021) | 68.68 | 61.85 | 57.43 | 52.68 | 50.19 | 46.88 | 44.65 | 43.07 | 40.17 | 39.63 | 37.33 | -21.97 |
| CEC Zhang et al. (2021) | 75.85 | 71.94 | 68.50 | 63.50 | 62.43 | 58.27 | 57.73 | 55.81 | 54.83 | 53.52 | 52.28 | -7.02 |
| F2M Shi et al. (2021) | 77.13 | 73.92 | 70.27 | 66.37 | 64.34 | 61.69 | 60.52 | 59.38 | 57.15 | 56.94 | 55.89 | -3.41 |
| LIMIT Zhou et al. (2022) | 75.89 | 73.55 | **71.99** | **68.14** | **67.42** | **63.61** | 62.40 | 61.35 | 59.91 | 58.66 | 57.41 | -1.89 |
| MetaFSCIL Chi et al. (2022) | 75.90 | 72.41 | 68.78 | 64.78 | 62.96 | 59.99 | 58.30 | 56.85 | 54.78 | 53.82 | 52.64 | -6.66 |
| ALICE Peng et al. (2022) | 77.40 | 72.70 | 70.60 | 67.20 | 65.90 | 63.40 | **62.90** | **61.90** | **60.50** | **60.60** | **60.10** | **-0.02** |
| Entropy-Reg Liu et al. (2022) | 75.90 | 72.14 | 68.64 | 63.76 | 62.58 | 59.11 | 57.82 | 55.89 | 54.92 | 53.58 | 52.39 | -6.91 |
| FSLL Mazumder et al. (2021) | 72.77 | 69.33 | 65.51 | 62.66 | 61.10 | 58.65 | 57.78 | 57.26 | 55.59 | 55.39 | 54.21 | -6.87 |
| FSLL+SS Mazumder et al. (2021) | 75.63 | 71.81 | 68.16 | 64.32 | 62.61 | 60.10 | 58.82 | 58.70 | 56.45 | 56.41 | 55.82 | -5.26 |
| HardNet, $c = 88\%$ | 76.89 | 73.40 | 69.77 | 66.15 | 64.00 | 60.98 | 59.56 | 58.05 | 56.05 | 55.84 | 55.20 | -4.10 |
| HardNet, $c = 90\%$ | 77.23 | 73.62 | 70.20 | 66.36 | 64.32 | 61.40 | 59.86 | 58.28 | 56.36 | 55.88 | 55.30 | -4.00 |
| HardNet, $c = 93\%$ | 77.76 | 73.97 | 70.41 | 66.60 | 64.47 | 61.35 | 59.80 | 58.18 | 56.17 | 55.73 | 55.18 | -4.12 |
| SoftNet, $c = 88\%$ | **78.14** | **74.61** | 71.28 | 67.46 | 65.14 | 62.39 | 60.84 | 59.17 | 57.41 | 57.12 | 56.64 | -2.66 |
| SoftNet, $c = 90\%$ | 78.07 | 74.58 | 71.37 | 67.54 | 65.37 | 62.60 | 61.07 | 59.37 | 57.53 | 57.21 | 56.75 | -2.55 |
| SoftNet, $c = 93\%$ | 78.11 | 74.51 | 71.14 | 62.27 | 65.14 | 62.27 | 60.77 | 59.03 | 57.13 | 56.77 | 56.28 | -3.02 |

This point makes it difficult to train AANet on a few samples even though the performance at session 1 is comparable with SoftNet as shown in Table 8.

Table 8: Classification accuracy of ResNet18 on CIFAR-100 for 5-way 5-shot incremental learning with the same class split as in TOPIC (Cheraghian et al., 2021). * denotes the results reported from Shi et al. (2021). † represents our reproduced results.

| Method | sessions | | | | | | | | | The gap with cRT |
|---|---|---|---|---|---|---|---|---|---|---|
| | 1 | 2 | 3 | 4 | 5 | 6 | 7 | 8 | 9 | |
| cRT Shi et al. (2021)* | 72.28 | 69.58 | 65.16 | 61.41 | 58.83 | 55.87 | 53.28 | 51.38 | 49.51 | |
| AANet Liu et al. (2021)† | 79.05 | 67.52 | 62.33 | 56.10 | 51.92 | 45.92 | 45.92 | 48.38 | 47.21 | -2.30 |
| HardNet, $c = 50\%$ | 78.35 | 74.12 | 70.13 | 65.88 | 62.74 | 59.56 | 57.98 | 56.31 | 54.32 | +4.81 |
| HardNet, $c = 80\%$ | 79.27 | 75.38 | 71.11 | 66.68 | 63.32 | 60.06 | 58.16 | 56.40 | 54.31 | +4.80 |
| HardNet, $c = 90\%$ | 79.22 | 74.77 | 70.89 | 66.41 | 62.90 | 59.48 | 58.10 | 56.13 | 53.92 | +4.41 |
| SoftNet, $c = 50\%$ | 79.88 | 75.54 | 71.64 | 67.47 | 64.45 | 61.09 | 59.07 | 57.29 | **55.33** | **+5.82** |
| SoftNet, $c = 80\%$ | **80.33** | **76.23** | **72.19** | **67.83** | **64.64** | **61.39** | **59.32** | **57.37** | 54.94 | +5.43 |
| SoftNet, $c = 90\%$ | 79.97 | 75.75 | 71.76 | 67.36 | 64.09 | 60.91 | 59.07 | 56.94 | 54.76 | +5.25 |

# E    LIMITATIONS AND FUTURE WORKS

Our method employs two sets of subnetworks. One is the major subnetworks, whereas the other is minor subnets. Since the former serve their duty to retain the base session knowledge, once the major subnetwork is tuned, there could be a potential loss of previously acquired knowledge. Furthermore, we explicitly divide SoftNet by the magnitude criterion. As a result, when SoftNet parameters are exposed, the essential parameters will be vulnerable to intentional attacks. It could result in the leak of knowledge maintained by SoftNet. First, to avoid tunning the major subnetwork issue, the new session learner should know the model sparsity for maintaining base session knowledge. Second, to address the leaking information issue, the binary mask should be encoded by a compression method to reduce model capacity and protect the privacy of task knowledge. Moreover, in FSCIL tasks, SoftNet alleviates overfitting issues while effectively maintaining base-session performance. In future work, we consider expanding the model parameters to acquire a long sequence of incoming new class knowledge depending on the data or task size, i.e., CIL tasks.

Table 9: Classification accuracy of ResNet18 on CIFAR-100 for 5-way 5-shot incremental learning. Underbar denotes the comparable results with baseline. * denotes the results reported from Shi et al. (2021).

| Method | sessions | | | | | | | | | The gap with cRT |
|---|---|---|---|---|---|---|---|---|---|---|
| | 1 | 2 | 3 | 4 | 5 | 6 | 7 | 8 | 9 | |
| cRT Shi et al. (2021) | 65.18 | 63.89 | 60.20 | 57.23 | 53.71 | 50.39 | 48.77 | 47.29 | 45.28 | - |
| Joint-training Shi et al. (2021) | 65.18 | 61.45 | 57.36 | 53.68 | 50.84 | 47.33 | 44.79 | 42.62 | 40.08 | -5.20 |
| Baseline Shi et al. (2021) | 65.18 | 61.67 | 58.61 | 55.11 | 51.86 | 49.43 | 47.60 | 45.64 | 43.83 | -1.45 |
| iCaRL Rebuffi et al. (2017)* | 66.52 | 57.26 | 54.27 | 50.62 | 47.33 | 44.99 | 43.14 | 41.16 | 39.49 | -5.79 |
| Rebalance Hou et al. (2019)* | 66.66 | 61.42 | 57.29 | 53.02 | 48.85 | 45.68 | 43.06 | 40.56 | 38.35 | -6.93 |
| FSLL Mazumder et al. (2021)* | 65.18 | 56.24 | 54.55 | 51.61 | 49.11 | 47.27 | 45.35 | 43.95 | 42.22 | -3.08 |
| iCaRL Rebuffi et al. (2017) | 64.10 | 53.28 | 41.69 | 34.13 | 27.93 | 25.06 | 20.41 | 15.48 | 13.73 | -31.55 |
| Rebalance Hou et al. (2019) | 64.10 | 53.05 | 43.96 | 36.97 | 31.61 | 26.73 | 21.23 | 16.78 | 13.54 | -31.74 |
| TOPIC Cheraghian et al. (2021) | 64.10 | 55.88 | 47.07 | 45.16 | 40.11 | 36.38 | 33.96 | 31.55 | 29.37 | -15.91 |
| F2M Shi et al. (2021) | 64.71 | 62.05 | 59.01 | 55.58 | 52.55 | 49.96 | 48.08 | 46.28 | 44.67 | -0.61 |
| FSLL Mazumder et al. (2021) | 64.10 | 55.85 | 51.71 | 48.59 | 45.34 | 43.25 | 41.52 | 39.81 | 38.16 | -7.12 |
| HardNet, $c = 10\%$ | 28.97 | 27.71 | 26.08 | 24.68 | 23.34 | 22.02 | 21.12 | 20.48 | 19.50 | -25.78 |
| HardNet, $c = 20\%$ | 37.42 | 35.29 | 33.22 | 31.32 | 29.51 | 27.80 | 26.54 | 25.28 | 24.16 | -21.12 |
| HardNet, $c = 30\%$ | 55.47 | 52.37 | 49.38 | 46.53 | 43.88 | 41.50 | 39.58 | 37.82 | 36.06 | -9.22 |
| HardNet, $c = 40\%$ | 57.52 | 53.85 | 50.62 | 47.74 | 44.90 | 42.64 | 40.76 | 38.95 | 37.07 | -8.21 |
| HardNet, $c = 50\%$ | 64.80 | 60.77 | 56.95 | 53.53 | 50.40 | 47.82 | 45.93 | 43.95 | 41.91 | -3.37 |
| HardNet, $c = 60\%$ | 66.72 | 62.21 | 58.14 | 54.60 | 51.47 | 48.86 | 46.67 | 44.67 | 42.66 | -2.62 |
| HardNet, $c = 70\%$ | 68.27 | 63.52 | 59.45 | 55.89 | 52.91 | 50.30 | 48.27 | 46.25 | 44.22 | -1.06 |
| HardNet, $c = 80\%$ | 69.65 | 64.60 | 60.59 | 56.93 | 53.60 | 50.80 | 48.69 | 46.69 | 44.63 | -0.65 |
| HardNet, $c = 90\%$ | 70.85 | 65.84 | 61.59 | 57.92 | 54.65 | 51.90 | 49.79 | 47.66 | 45.47 | +0.19 |
| HardNet, $c = 93\%$ | 71.22 | 66.20 | 62.00 | 58.34 | 55.04 | 52.34 | 50.22 | 48.07 | 46.04 | +0.76 |
| HardNet, $c = 95\%$ | 71.73 | 66.31 | 62.17 | 58.44 | 54.98 | 52.20 | 50.17 | 47.97 | 45.87 | +0.59 |
| HardNet, $c = 97\%$ | 71.85 | 66.48 | 62.29 | 58.62 | 55.36 | 52.55 | 50.60 | 48.43 | 46.22 | +0.94 |
| HardNet, $c = 99\%$ | 71.95 | 66.83 | 62.75 | 59.09 | 55.92 | 53.03 | 50.78 | 48.52 | 46.31 | +1.03 |
| SoftNet, $c = 10\%$ | 60.77 | 57.02 | 53.62 | 50.51 | 47.67 | 45.14 | 43.32 | 41.6 | 39.58 | -5.70 |
| SoftNet, $c = 20\%$ | 64.67 | 60.69 | 57.15 | 53.77 | 50.76 | 48.28 | 46.24 | 44.23 | 42.31 | -2.97 |
| SoftNet, $c = 30\%$ | 67.00 | 62.18 | 58.22 | 54.69 | 51.82 | 49.12 | 47.13 | 44.98 | 42.44 | -2.84 |
| SoftNet, $c = 40\%$ | 67.50 | 63.11 | 59.29 | 55.61 | 52.53 | 49.85 | 47.85 | 45.84 | 43.85 | -1.43 |
| SoftNet, $c = 50\%$ | 69.20 | 64.18 | 60.01 | 56.43 | 53.11 | 50.62 | 48.60 | 46.51 | 44.61 | -0.67 |
| SoftNet, $c = 60\%$ | 69.15 | 63.68 | 59.54 | 56.05 | 52.72 | 50.10 | 48.20 | 46.18 | 44.15 | -1.13 |
| SoftNet, $c = 70\%$ | 70.92 | 65.16 | 61.00 | 57.25 | 54.09 | 51.37 | 49.29 | 47.03 | 44.90 | -0.38 |
| SoftNet, $c = 80\%$ | 70.38 | 65.04 | 60.94 | 57.26 | 54.13 | 51.58 | 49.52 | 47.36 | 45.16 | -0.12 |
| SoftNet, $c = 90\%$ | 72.25 | 66.82 | 62.63 | 58.98 | 55.64 | 52.77 | 50.71 | 48.42 | 46.15 | +0.87 |
| SoftNet, $c = 93\%$ | 71.38 | 65.93 | 61.89 | 58.20 | 54.87 | 51.83 | 49.82 | 47.57 | 45.47 | +0.19 |
| SoftNet, $c = 95\%$ | 72.23 | 66.94 | 62.56 | 58.84 | 55.65 | 52.74 | 50.61 | 48.47 | 46.27 | +0.99 |
| SoftNet, $c = 97\%$ | 70.88 | 65.72 | 61.38 | 57.88 | 54.63 | 51.82 | 49.57 | 47.30 | 45.19 | -0.09 |
| SoftNet, $c = 99\%$ | **72.62** | **67.31** | **63.05** | **59.39** | **56.00** | **53.23** | **51.06** | **48.83** | **46.63** | **+1.35** |

Table 10: Classification accuracy of ResNet18 v.s. ResNet20 on CIFAR-100 for 5-way 5-shot FSCIL with varying capacity $c$. Underbar denotes the comparable results with baseline. $^*$ denotes results reported from Shi et al. (2021).

| Method | sessions | | | | | | | | | The gap with cRT |
|---|---|---|---|---|---|---|---|---|---|---|
| | 1 | 2 | 3 | 4 | 5 | 6 | 7 | 8 | 9 | |
| ResNet18, cRT Shi et al. (2021) | 65.18 | 63.89 | 60.20 | 57.23 | 53.71 | 50.39 | 48.77 | 47.29 | 45.28 | - |
| ResNet18, Joint-training Shi et al. (2021) | 65.18 | 61.45 | 57.36 | 53.68 | 50.84 | 47.33 | 44.79 | 42.62 | 40.08 | -5.20 |
| ResNet18, Baseline Shi et al. (2021) | 65.18 | 61.67 | 58.61 | 55.11 | 51.86 | 49.43 | 47.60 | 45.64 | 43.83 | -1.45 |
| ResNet18, HardNet, $c = 10\%$ | 28.97 | 27.71 | 26.08 | 24.68 | 23.34 | 22.02 | 21.12 | 20.48 | 19.50 | -25.78 |
| ResNet18, HardNet, $c = 20\%$ | 37.42 | 35.29 | 33.22 | 31.32 | 29.51 | 27.80 | 26.54 | 25.28 | 24.16 | -21.12 |
| ResNet18, HardNet, $c = 30\%$ | 55.47 | 52.37 | 49.38 | 46.53 | 43.88 | 41.50 | 39.58 | 37.82 | 36.06 | -9.22 |
| ResNet18, HardNet, $c = 40\%$ | 57.52 | 53.85 | 50.62 | 47.74 | 44.90 | 42.64 | 40.76 | 38.95 | 37.07 | -8.21 |
| ResNet18, HardNet, $c = 50\%$ | 64.80 | 60.77 | 56.95 | 53.53 | 50.40 | 47.82 | 45.93 | 43.95 | 41.91 | -3.37 |
| ResNet18, HardNet, $c = 60\%$ | 66.72 | 62.21 | 58.14 | 54.60 | 51.47 | 48.86 | 46.67 | 44.67 | 42.66 | -2.62 |
| ResNet18, HardNet, $c = 70\%$ | 68.27 | 63.52 | 59.45 | 55.89 | 52.91 | 50.30 | 48.27 | 46.25 | 44.22 | -1.06 |
| ResNet18, HardNet, $c = 80\%$ | 69.65 | 64.60 | 60.59 | 56.93 | 53.60 | 50.80 | 48.69 | 46.69 | 44.63 | -0.65 |
| ResNet18, HardNet, $c = 90\%$ | 70.85 | 65.84 | 61.59 | 57.92 | 54.65 | 51.90 | 49.79 | 47.66 | 45.47 | +0.19 |
| ResNet18, HardNet, $c = 93\%$ | 71.22 | 66.20 | 62.00 | 58.34 | 55.04 | 52.34 | 50.22 | 48.07 | 46.04 | +0.76 |
| ResNet18, HardNet, $c = 95\%$ | 71.73 | 66.31 | 62.17 | 58.44 | 54.98 | 52.20 | 50.17 | 47.97 | 45.87 | +0.59 |
| ResNet18, HardNet, $c = 97\%$ | 71.85 | 66.48 | 62.29 | 58.62 | 55.36 | 52.55 | 50.60 | 48.43 | 46.22 | +0.94 |
| ResNet18, HardNet, $c = 99\%$ | 71.95 | 66.83 | 62.75 | 59.09 | 55.92 | 53.03 | 50.78 | 48.52 | 46.31 | +1.03 |
| ResNet18, SoftNet, $c = 10\%$ | 60.77 | 57.02 | 53.62 | 50.51 | 47.67 | 45.14 | 43.32 | 41.6 | 39.58 | -5.7 |
| ResNet18, SoftNet, $c = 20\%$ | 64.67 | 60.69 | 57.15 | 53.77 | 50.76 | 48.28 | 46.24 | 44.23 | 42.31 | -2.97 |
| ResNet18, SoftNet, $c = 30\%$ | 67.00 | 62.18 | 58.22 | 54.69 | 51.82 | 49.12 | 47.13 | 44.98 | 42.44 | -2.84 |
| ResNet18, SoftNet, $c = 40\%$ | 67.50 | 63.11 | 59.29 | 55.61 | 52.53 | 49.85 | 47.85 | 45.84 | 43.85 | -1.43 |
| ResNet18, SoftNet, $c = 50\%$ | 69.20 | 64.18 | 60.01 | 56.43 | 53.11 | 50.62 | 48.60 | 46.51 | 44.61 | -0.67 |
| ResNet18, SoftNet, $c = 60\%$ | 69.15 | 63.68 | 59.54 | 56.05 | 52.72 | 50.10 | 48.20 | 46.18 | 44.15 | -1.13 |
| ResNet18, SoftNet, $c = 70\%$ | 70.92 | 65.16 | 61.00 | 57.25 | 54.09 | 51.37 | 49.29 | 47.03 | 44.90 | -0.38 |
| ResNet18, SoftNet, $c = 80\%$ | 70.38 | 65.04 | 60.94 | 57.26 | 54.13 | 51.58 | 49.52 | 47.36 | 45.16 | -0.12 |
| ResNet18, SoftNet, $c = 90\%$ | 72.25 | 66.82 | 62.63 | 58.98 | 55.64 | 52.77 | 50.71 | 48.42 | 46.15 | +0.87 |
| ResNet18, SoftNet, $c = 93\%$ | 71.38 | 65.93 | 61.89 | 58.20 | 54.87 | 51.83 | 49.82 | 47.57 | 45.47 | +0.19 |
| ResNet18, SoftNet, $c = 95\%$ | 72.23 | 66.94 | 62.56 | 58.84 | 55.65 | 52.74 | 50.61 | 48.47 | 46.27 | +0.99 |
| ResNet18, SoftNet, $c = 97\%$ | 70.88 | 65.72 | 61.38 | 57.88 | 54.63 | 51.82 | 49.57 | 47.30 | 45.19 | -0.09 |
| ResNet18, SoftNet, $c = 99\%$ | **72.62** | **67.31** | **63.05** | **59.39** | **56.00** | **53.23** | **51.06** | **48.83** | **46.63** | **+1.35** |
| ResNet20, HardNet, $c = 10\%$ | 53.50 | 50.52 | 47.79 | 45.11 | 42.57 | 40.38 | 38.88 | 37.43 | 35.69 | -9.59 |
| ResNet20, HardNet, $c = 20\%$ | 61.83 | 58.24 | 55.17 | 51.92 | 49.07 | 46.68 | 45.02 | 43.31 | 41.40 | -3.88 |
| ResNet20, HardNet, $c = 30\%$ | 66.68 | 62.81 | 59.54 | 56.12 | 53.21 | 50.52 | 48.85 | 46.93 | 44.84 | -0.44 |
| ResNet20, HardNet, $c = 40\%$ | 68.31 | 64.17 | 60.72 | 57.19 | 54.26 | 51.79 | 50.13 | 48.01 | 45.93 | +0.65 |
| ResNet20, HardNet, $c = 50\%$ | 70.73 | 66.56 | 63.10 | 59.59 | 56.58 | 53.87 | 51.86 | 49.72 | 47.46 | +2.18 |
| ResNet20, HardNet, $c = 60\%$ | 71.90 | 67.64 | 63.84 | 60.22 | 57.06 | 54.30 | 52.53 | 50.53 | 48.34 | +3.06 |
| ResNet20, HardNet, $c = 70\%$ | 71.41 | 67.44 | 63.76 | 60.02 | 56.84 | 54.14 | 52.54 | 50.42 | 48.24 | +2.96 |
| ResNet20, HardNet, $c = 80\%$ | 71.97 | 67.65 | 63.93 | 60.14 | 57.12 | 54.44 | 52.72 | 50.67 | 48.43 | +3.15 |
| ResNet20, HardNet, $c = 90\%$ | 71.82 | 67.73 | 64.21 | 60.44 | 57.44 | 54.93 | 53.14 | 51.03 | 48.86 | +3.58 |
| ResNet20, HardNet, $c = 93\%$ | 72.28 | 67.99 | 64.48 | 60.83 | 57.64 | 55.16 | 53.27 | 51.11 | 48.93 | +3.65 |
| ResNet20, HardNet, $c = 95\%$ | 72.13 | 68.14 | 64.50 | 60.74 | 57.68 | 55.12 | 53.17 | 51.23 | 48.97 | +3.69 |
| ResNet20, HardNet, $c = 97\%$ | 71.90 | 67.81 | 64.11 | 60.24 | 57.14 | 54.41 | 52.74 | 50.71 | 48.67 | +3.39 |
| ResNet20, SoftNet, $c = 10\%$ | 53.13 | 49.73 | 46.85 | 44.01 | 41.54 | 39.45 | 37.84 | 36.30 | 34.62 | -10.66 |
| ResNet20, SoftNet, $c = 20\%$ | 60.15 | 56.25 | 53.26 | 50.16 | 47.46 | 45.23 | 43.75 | 41.84 | 39.90 | -5.38 |
| ResNet20, SoftNet, $c = 30\%$ | 64.65 | 59.99 | 56.65 | 53.45 | 50.42 | 48.02 | 46.48 | 44.61 | 42.43 | -2.85 |
| ResNet20, SoftNet, $c = 40\%$ | 66.77 | 62.55 | 59.35 | 55.63 | 52.70 | 50.21 | 48.54 | 46.60 | 44.50 | -0.78 |
| ResNet20, SoftNet, $c = 50\%$ | 68.20 | 64.21 | 60.78 | 57.15 | 54.20 | 51.53 | 49.84 | 47.92 | 45.84 | +0.56 |
| ResNet20, SoftNet, $c = 60\%$ | 68.63 | 64.76 | 61.00 | 57.53 | 54.56 | 52.11 | 50.34 | 48.41 | 46.30 | +1.02 |
| ResNet20, SoftNet, $c = 70\%$ | 70.38 | 66.16 | 62.63 | 58.93 | 55.81 | 53.11 | 51.38 | 49.29 | 47.08 | +1.80 |
| ResNet20, SoftNet, $c = 80\%$ | 70.87 | 66.47 | 62.85 | 59.28 | 56.27 | 53.84 | 52.01 | 50.16 | 48.01 | +2.73 |
| ResNet20, SoftNet, $c = 90\%$ | 72.63 | **68.60** | **64.96** | **61.25** | **57.98** | **55.32** | **53.48** | **51.46** | **49.20** | **+3.92** |
| ResNet20, SoftNet, $c = 93\%$ | 72.53 | 68.45 | 64.59 | 60.80 | 57.48 | 54.91 | 53.19 | 51.03 | 48.83 | +3.55 |
| ResNet20, SoftNet, $c = 95\%$ | 72.58 | 68.30 | 64.57 | 60.83 | 57.67 | 54.86 | 53.11 | 51.21 | 49.06 | +3.78 |
| ResNet20, SoftNet, $c = 97\%$ | **72.80** | 68.46 | 64.61 | 60.90 | 57.63 | 54.95 | 53.26 | 51.12 | 48.95 | +3.67 |
| ResNet20, SoftNet, $c = 99\%$ | 71.78 | 67.79 | 63.86 | 60.07 | 57.05 | 54.32 | 52.34 | 50.28 | 48.11 | +2.83 |

Table 11: Classification accuracy of ResNet18 on miniImageNet for 5-way 5-shot incremental learning. Underbar denotes the comparable results with baseline. ∗ denotes results reported from Shi et al. (2021).

| Method | sessions | | | | | | | | | The gap with cRT |
|---|---|---|---|---|---|---|---|---|---|---|
| | 1 | 2 | 3 | 4 | 5 | 6 | 7 | 8 | 9 | |
| cRT Shi et al. (2021) | 67.30 | 64.15 | 60.59 | 57.32 | 54.22 | 51.43 | 48.92 | 46.78 | 44.85 | - |
| Joint-training Shi et al. (2021) | 67.30 | 62.34 | 57.79 | 54.08 | 50.93 | 47.65 | 44.64 | 42.61 | 40.29 | -4.56 |
| Baseline Shi et al. (2021) | 67.30 | 63.18 | 59.62 | 56.33 | 53.28 | 50.50 | 47.96 | 45.85 | 43.88 | -0.97 |
| iCaRL Rebuffi et al. (2017)∗ | 67.35 | 59.91 | 55.64 | 52.60 | 49.43 | 46.73 | 44.13 | 42.17 | 40.29 | -4.56 |
| Rebalance Hou et al. (2019)∗ | 67.91 | 63.11 | 58.75 | 54.83 | 50.68 | 47.11 | 43.88 | 41.19 | 38.72 | -6.13 |
| FSLL Mazumder et al. (2021)∗ | 67.30 | 59.81 | 57.26 | 54.57 | 52.05 | 49.42 | 46.95 | 44.94 | 42.87 | -1.11 |
| iCaRL Rebuffi et al. (2017) | 61.31 | 46.32 | 42.94 | 37.63 | 30.49 | 24.00 | 20.89 | 18.80 | 17.21 | -27.64 |
| Rebalance Hou et al. (2019) | 61.31 | 47.80 | 39.31 | 31.91 | 25.68 | 21.35 | 18.67 | 17.24 | 14.17 | -30.68 |
| TOPIC Cheraghian et al. (2021) | 61.31 | 50.09 | 45.17 | 41.16 | 37.48 | 35.52 | 32.19 | 29.46 | 24.42 | -20.43 |
| IDLVQ-C Chen and Lee (2020) | 64.77 | 59.87 | 55.93 | 52.62 | 49.88 | 47.55 | 44.83 | 43.14 | 41.84 | -3.01 |
| F2M Shi et al. (2021) | 67.28 | 63.80 | 60.38 | 57.06 | 54.08 | 51.39 | 48.82 | 46.58 | 44.65 | -0.20 |
| FSLL Mazumder et al. (2021) | 66.48 | 61.75 | 58.16 | 54.16 | 51.10 | 48.53 | 46.54 | 44.20 | 42.28 | -2.57 |
| HardNet, $c = 10\%$ | 9.70 | 8.46 | 8.13 | 7.67 | 7.35 | 6.78 | 6.55 | 6.3 | 5.8 | -39.05 |
| HardNet, $c = 20\%$ | 24.95 | 22.20 | 20.66 | 19.39 | 18.32 | 17.41 | 16.64 | 15.79 | 15.28 | -29.57 |
| HardNet, $c = 30\%$ | 44.08 | 40.66 | 38.24 | 36.06 | 33.65 | 31.84 | 30.25 | 29.01 | 28.02 | -16.83 |
| HardNet, $c = 40\%$ | 37.05 | 34.94 | 32.25 | 30.72 | 29.09 | 27.53 | 25.84 | 24.76 | 23.71 | -21.14 |
| HardNet, $c = 50\%$ | 65.13 | 60.37 | 56.12 | 53.17 | 50.17 | 47.74 | 45.34 | 43.35 | 42.13 | -2.72 |
| HardNet, $c = 60\%$ | 73.32 | 67.77 | 63.46 | 60.13 | 57.13 | 50.36 | 47.88 | 46.05 | 44.59 | -0.26 |
| HardNet, $c = 70\%$ | 71.75 | 66.66 | 62.19 | 58.85 | 55.74 | 52.82 | 50.14 | 48.45 | 47.01 | +2.16 |
| HardNet, $c = 80\%$ | 69.73 | 64.46 | 60.42 | 57.09 | 54.09 | 51.18 | 48.76 | 46.81 | 45.66 | +0.81 |
| HardNet, $c = 90\%$ | 64.68 | 59.80 | 55.70 | 52.82 | 50.01 | 47.30 | 45.17 | 43.34 | 42.09 | -2.76 |
| HardNet, $c = 93\%$ | 67.17 | 61.74 | 57.53 | 54.43 | 51.52 | 48.86 | 46.42 | 44.68 | 43.43 | -1.42 |
| HardNet, $c = 95\%$ | 64.72 | 60.13 | 56.05 | 53.25 | 50.20 | 47.62 | 45.11 | 43.40 | 42.33 | -2.52 |
| HardNet, $c = 97\%$ | 63.92 | 58.85 | 55.12 | 52.16 | 49.44 | 46.78 | 44.48 | 42.68 | 41.52 | -3.33 |
| HardNet, $c = 99\%$ | 67.28 | 62.17 | 58.06 | 55.05 | 52.01 | 49.12 | 46.92 | 45.07 | 43.90 | -0.95 |
| SoftNet, $c = 10\%$ | 62.75 | 58.26 | 53.80 | 50.82 | 47.68 | 44.86 | 42.05 | 39.86 | 38.15 | -6.70 |
| SoftNet, $c = 20\%$ | 68.33 | 62.76 | 58.60 | 55.25 | 52.07 | 49.36 | 46.48 | 44.21 | 42.52 | -2.33 |
| SoftNet, $c = 30\%$ | 72.20 | 66.92 | 62.56 | 59.16 | 56.07 | 53.10 | 50.59 | 48.46 | 47.03 | +2.18 |
| SoftNet, $c = 40\%$ | 72.58 | 67.34 | 62.91 | 59.65 | 56.72 | 54.00 | 51.46 | 49.39 | 48.11 | +3.26 |
| SoftNet, $c = 50\%$ | 72.83 | 67.23 | 62.82 | 59.41 | 56.44 | 53.55 | 50.92 | 48.99 | 47.60 | +2.75 |
| SoftNet, $c = 60\%$ | 73.83 | 67.78 | 63.46 | 60.21 | 57.27 | 54.42 | 51.74 | 49.94 | 48.57 | +3.72 |
| SoftNet, $c = 70\%$ | 75.15 | 69.06 | 64.79 | 61.40 | 58.38 | 55.49 | 52.87 | 50.89 | 49.69 | +4.84 |
| SoftNet, $c = 80\%$ | 76.63 | 70.13 | 65.92 | 62.52 | **59.49** | **56.56** | 53.71 | 51.72 | **50.48** | **+5.63** |
| SoftNet, $c = 90\%$ | 77.00 | **70.38** | 65.94 | 62.45 | 59.32 | 56.25 | **53.76** | **51.75** | 50.39 | +5.54 |
| SoftNet, $c = 93\%$ | 73.97 | 67.39 | 63.35 | 59.90 | 56.89 | 54.18 | 51.61 | 49.71 | 48.45 | +3.60 |
| SoftNet, $c = 95\%$ | 76.22 | 69.64 | 65.22 | 61.91 | 58.84 | 55.75 | 53.07 | 51.18 | 49.84 | +4.99 |
| SoftNet, $c = 97\%$ | **77.17** | 70.32 | **66.15** | **62.55** | 59.48 | 56.46 | 53.71 | 51.68 | 50.24 | +5.39 |
| SoftNet, $c = 99\%$ | 76.80 | 69.79 | 65.44 | 62.01 | 58.87 | 55.94 | 53.21 | 51.25 | 50.04 | +5.19 |

