# OpenReview forum: "On the Soft-Subnetwork for Few-Shot Class Incremental Learning"
_ICLR.cc/2023/Conference — ICLR 2023 poster_

### Official Review · Reviewer_5XQs · 2022-10-24

**Confidence:** 5
**Correctness:** 3
**Technical Novelty And Significance:** 2
**Empirical Novelty And Significance:** 2
**Recommendation:** 5

**Clarity, Quality, Novelty And Reproducibility:**

The clarity of this work is good, but the novelty and experimental performance are limited, which impedes the quality of this paper.  Details are stated in the Weaknesses section.

**Strength And Weaknesses:**

Strengths:

1). The paper is well present. In particular, Fig 1 and Algorithm 1 are clear and informative.

2). The method is simple and easy to follow.

Weaknesses:

1). From the comparison experiments, it is hard to distinguish whether the proposed method has a good performance since there are plenty of works for few-shot class-incremental learning that the paper fails to have a comparison with.

- Few-shot incremental learning with continually evolved classifiers. (CVPR ‘21)
- Few-shot class incremental learning by sampling multi-phase tasks. (TPAMI)
- Subspace regularizers for few-shot class incremental learning. (ICLR ‘22)
- Metafscil: A meta-learning approach for few-shot class incremental learning. (CVPR ‘22)
- Constrained few-shot class-incremental learning. (CVPR ‘22)
- Few-shot class incremental learning via entropy-regularized data-free replay. (ECCV ‘22)
- Few-shot class-incremental learning from an open-set perspective. (ECCV ‘22)

As an ICLR 23 submission, most of the baseline methods in this paper are from 2 years ago. An important baseline in FSCIL, CEC (Zhang et al., CVPR 2021), is even not compared with. I want to note that the above-mentioned methods seem to outperform the proposed method in this paper by a large margin.

2). The proposed method seems to have a better result than F2M in the final session, but with a much better results in the session 1, which may indicate that the proposed method cannot prevent the network from forgetting, which is opposite from what is claimed in the introduction.

3). The proposed method sounds not that new to me. I found the authors fail to give a reference for AANet (Adaptive Aggregation Networks for Class-Incremental Learning, CVPR ‘21). However, from my own perspective, the proposed method shares a similar motivation with AANet since both want to alleviate the catastrophic forgetting by fixing some parameters inside the network. Though CIL and FSCIL are different benchmarks, AANet seems to perform better and have a capability of preventing catastrophic forgetting. The authors should discuss the comparison with AANet in details.


-------After rebuttal-----

The authors offer detailed response and clarification. I am happy to increase the score, but I sitll have the concern about the similar motivation to AANet.


**Summary Of The Paper:**

The paper proposes a Regularized Lottery Ticket Hypothesis inspired network to deal with the well-known catastrophic forgetting problem. By only updating partial weights in the network, the proposed method claims to have a good performance on both the classes in previous and current sessions.

**Summary Of The Review:**

Despite good paper writing, the novelty and the experimental performance are the largest issues of this paper. Most of the baseline methods are from 2 years ago. An important baseline in FSCIL, CEC (Zhang et al., CVPR 2021), is even not compared with.

See weaknesses for details.

---

> ### Author Response · Authors · 2022-11-12
> **Outline of our responses**
>
> Dear Reviewer 5XQs,
>
> Thank reviewer 5XQs for constructive feedback. We will address all concerns about additional experiments, misleading points in the introduction, and the difference between SoftNet and AANet. After completing the additional experiments, we would like to prepare our responses to the reviewer's comments as follows:
>
> - Additional experiments: we will report comparing SoftNet with seven different current methods on CIFAR100, miniImageNet, and CUB200.
>
> - Misleading points: we will revise the keyword in the introduction, i.e., SoftNet alleviates the model's base session knowledge from forgetting.
>
> - The difference between SoftNet and AANet: we will depict the main differences between SoftNet and AANet in terms of the objective, model structures, model fixing methods, and pros/cons.

---

> > ### Author Response · Authors · 2022-11-16
> > **Table 5. Comparisons of AANet and SOftNet on CIFAR100**
> >
> > | **Method**                     | 1              | 2              | 3              | 4              | 5              | 6              | 7              | 8              | 9              | **The gap with cRT** |
> > |-------------------------------------|----------------|----------------|----------------|----------------|----------------|----------------|----------------|----------------|----------------|---------------------------|
> > | cRT Shi et al. (2021)$^\ast$        | 72.28          | 69.58          | 65.16          | 61.41          | 58.83          | 55.87          | 53.28          | 51.38          | 49.51          |
> > | AANet Liu et al. (2021)$^\dagger$   | 79.05          | 67.52          | 62.33          | 56.10          | 51.92          | 45.92          | 45.92          | 48.38          | 47.21          | -2.30                     |
> > | SoftNet (best)                      | **79.88**      | **75.54**      | **71.64**      | **67.47**      | **64.45**      | **61.09**      | **59.07**      | **57.29**      | **55.33**      | **+5.82**                 |

---

> > ### Author Response · Authors · 2022-11-16
> > **Table 4. Comparisons of AANet and SoftNet**
> >
> > - Regarding the comparison with AANet as shown in Table 4, we argue that SoftNet and AANet both aim to alleviate catastrophic forgetting in FSCIL and CIL, respectively. However, our work specifically aims to: (1) Find regularized subnetwork using only a single dense network and (2) Learn knowledge on incoming data which belongs to a new class while maintaining base knowledge leveraged by the acquired soft subnetwork. The following Table highlights major differences concerning the main objective, structures, and pros \& cons between SoftNet and AANet.
> >
> > |                               | AANet (CVPR2021)                                                                                                                                                                                                                                                                                                                                                                                                                                                                      | SoftNet (Ours)                                                                                                                                                                                                                                                                                                                                                                                                                                                                                                                                                                         |
> > |-------------|----------|-----|
> > | Objective   | **(Multi-ResNets):** To maintain previously learned knowledge while simultaneously acquiring new knowledge from incoming novel classes                                                                                                                                                                                                                                                                                                                                                    | **(A Single ResNet):** To preserve knowledge within the base session and acquire new knowledge via regularized minor subnetworks using backbone with fixed sparsity $c$.                                                                                                                                                                                                                                                                                                                                                                                                                 |
> > | Model Structures | **Two residual blocks:** (1) Stable block for maintaining the knowledge of old classes and (2) Plastic block for learning new classes in Class Incremental Learning.                                                                                                                                                                                                                                                                                                   | **Single residual block:** (1) Major subnetwork $m_{major}$ for maintaining knowledge in base session,  and (2) Minor subnetwork $m_{minor}$ for learning incoming novel classes in FSCIL.                                                                                                                                                                                                                                                                         |
> > | HOW TO Fix Model Parameters | AANet adapts a bi-level Optimization scheme, where the first optimization is regarding updates of stable blocks and plastic blocks; the second optimization to obtain aggregation weights for the two residual blocks                                                                                                                                                                                                                          | SofNet adaptively fixes major and minor subnetworks during the base session to obtain the regularized soft-subnetwork.                                                                                                                                                                                                                                                      |
> > | Pros and Cons | **Pros:** AANet can learn CSIL tasks involving long-sequence and balanced data.  **Cons:** AANet suffers from overfitting in FSCIL since it uses and updates whole parameters, as shown in Table 5. | **Pros:** SoftNet alleviates overfitting issues, while effectively maintaining base-session performance, as shown in Table 5.  **Cons:** In future work, we consider expanding the model parameters to acquire a long sequence of incoming new class knowledge depending on the data or task size. |

---

> > ### Author Response · Authors · 2022-11-16
> > **Correction of misleading points and the trade-off in terms of forgetting.**
> >
> > - Instead of preventing the network from forgetting, our proposed SoftNet effectively alleviates the catastrophic forgetting issued in the introduction. Experiment results showed SoftNet and various models' tendency to forget as the session training proceeds. In the appendix, we also pointed out the base and incoming new session performance trade-off between our proposed HardNet and SoftNet.

---

> > ### Author Response · Authors · 2022-11-16
> > **Table 3. CUB200 of TOPIC split**
> >
> > | **Method**                     | 1              | 2              | 3              | 4              | 5              | 6              | 7              | 8              | 9              | 10              | 11              | **The gap with cRT**|
> > |-----------------------------------------|----------------|----------------|--------------|----------------|----------------|----------------|----------------|----------------|----------------|----------------|----------------|----------------|
> > | cRT Shi et al. (2021)$^\ast$            | 77.16          | 74.41          | 71.31        | 68.08          | 65.57          | 63.08          | 62.44          | 61.29          | 60.12          | 59.85          | 59.30          | -              |
> > | TOPIC Cheraghian et al. (2021)          | 68.68          | 62.49          | 54.81        | 49.99          | 45.25          | 41.40          | 38.35          | 35.36          | 32.22          | 28.31          | 26.28          | -34.80         |
> > | SPPR Zhu et al. (2021)                  | 68.68          | 61.85          | 57.43        | 52.68          | 50.19          | 46.88          | 44.65          | 43.07          | 40.17          | 39.63          | 37.33          | -21.97         |
> > | CEC Zhang et al. (2021)                 | 75.85          | 71.94          | 68.50        | 63.50          | 62.43          | 58.27          | 57.73          | 55.81          | 54.83          | 53.52          | 52.28          | -7.02          |
> > | F2M Shi et al. (2021)                   | 77.13          | 73.92          | 70.27        | 66.37          | 64.34          | 61.69          | 60.52          | 59.38          | 57.15          | 56.94          | 55.89          | -3.41          |
> > | **LIMIT+Transformer**  Zhou et al. (2022)                | 75.89          | 73.55          | **71.99**    | **68.14**      | **67.42**      | **63.61**      | 62.40          | 61.35          | 59.91          | 58.66          | 57.41          | -1.89          |
> > | MetaFSCIL  Chi et al. (2022)            | 75.90          | 72.41          | 68.78        | 64.78          | 62.96          | 59.99          | 58.30          | 56.85          | 54.78          | 53.82          | 52.64          | -6.66          |
> > | **ALICE+Data Aug** Peng et al. (2022)                | 77.40          | 72.70          | 70.60        | 67.20          | 65.90          | 63.40          | **62.90**      | **61.90**      | **60.50**      | **60.60**      | **60.10**      | **-0.02**      |
> > | Entropy-Reg  Liu et al. (2022)          | 75.90          | 72.14          | 68.64        | 63.76          | 62.58          | 59.11          | 57.82          | 55.89          | 54.92          | 53.58          | 52.39          | -6.91          |
> > | FSLL Mazumder et al. (2021)             | 72.77          | 69.33          | 65.51        | 62.66          | 61.10          | 58.65          | 57.78          | 57.26          | 55.59          | 55.39          | 54.21          | -6.87          |
> > | FSLL+SS Mazumder et al. (2021)          | 75.63          | 71.81          | 68.16        | 64.32          | 62.61          | 60.10          | 58.82          | 58.70          | 56.45          | 56.41          | 55.82          | -5.26          |
> > | HardNet (best)                          | 77.23          | 73.62          | 70.20        | 66.36          | 64.32          | 61.40          | 59.86          | 58.28          | 56.36          | 55.88          | 55.30          | -4.00          |
> > | **SoftNet (best)**                          | **78.07**      | **74.58**      | 71.37        | 67.54          | 65.37          | 62.60          | 61.07          | 59.37          | 57.53          | 57.21          | 56.75          | -2.55          |

---

> > ### Author Response · Authors · 2022-11-16
> > **Table 2. MiniImageNet of TOPIC split**
> >
> > | **Method**                     | 1              | 2              | 3              | 4              | 5              | 6              | 7              | 8              | 9              | **The gap with cRT** |
> > |------------------------------------------------|--------------|----------------|----------------|----------------|----------------|----------------|----------------|----------------|----------------|----------------|
> > | cRT Shi et al. (2021)$^\ast$                   | 72.08        | 68.15          | 63.06          | 61.12          | 56.57          | 54.47          | 51.81          | 49.86          | 48.31          | -              |
> > | TOPIC Cheraghian et al. (2021)                 | 61.31        | 50.09          | 45.17          | 41.16          | 37.48          | 35.52          | 32.19          | 29.46          | 24.42          | -23.89         |
> > | IDLVQ-C Chen and Lee (2020)                    | 64.77        | 59.87          | 55.93          | 52.62          | 49.88          | 47.55          | 44.83          | 43.14          | 41.84          | -6.47          |
> > | CEC Zhang et al. (2021)                        | 72.00        | 66.83          | 62.97          | 59.43          | 56.70          | 53.73          | 51.19          | 49.24          | 47.63          | -0.68          |
> > | F2M Shi et al. (2021)                          | 72.05        | 67.47          | 63.16          | 59.70          | 56.71          | 53.77          | 51.11          | 49.21          | 47.84          | -0.43          |
> > | **LIMIT+Transformer**  Zhou et al. (2022)                       | 73.81        | 72.09          | 67.87          | 63.89          | 60.70          | 57.77          | 55.67          | 53.52          | 51.23          | +2.92          |
> > | MetaFSCIL  Chi et al. (2022)                   | 72.04        | 67.94          | 63.77          | 60.29          | 57.58          | 55.16          | 52.90          | 50.79          | 49.19          | +0.88          |
> > | **ALICE+Data Aug** Peng et al. (2022)                       | **80.60**    | 70.60          | 67.40          | 64.50          | 62.50          | 60.00          | 57.80          | **56.80**      | **55.70**      | +**7.39**      |
> > | C-FSCIL Hersche et al. (2022)                  | 76.40        | 71.14          | 66.46          | 63.29          | 60.42          | 57.46          | 54.78          | 53.11          | 51.41          | +3.10          |
> > | Entropy-Reg  Liu et al. (2022)                 | 71.84        | 67.12          | 63.21          | 59.77          | 57.01          | 53.95          | 51.55          | 49.52          | 48.21          | -0.10          |
> > | Subspace Reg. Akyürek et al. (2021)            | 80.37        | 71.69          | 66.94          | 62.53          | 58.90          | 55.00          | 51.94          | 49.76          | 46.79          | -1.52          |
> > | FSLL Mazumder et al. (2021)                    | 66.48        | 61.75          | 58.16          | 54.16          | 51.10          | 48.53          | 46.54          | 44.20          | 42.28          | -6.03          |
> > | FSLL+SS Mazumder et al. (2021)                 | 68.85        | 63.14          | 59.24          | 55.23          | 52.24          | 49.65          | 47.74          | 45.23          | 43.92          | -4.39          |
> > | HardNet (best)                                 | 79.17        | 73.05          | 69.16          | 65.43          | 62.61          | 59.31          | 56.73          | 54.69          | 53.47          | +5.16          |
> > | **SoftNet (best)**                                 | 79.77        | **75.08**      | **70.59**      | **66.93**      | **64.00**      | **61.00**      | **57.81**      | 55.81          | 54.68          | +6.37          |

---

> > ### Author Response · Authors · 2022-11-16
> > **Table 1. CIFAR100 of TOPIC split**
> >
> > | **Method**                     | 1              | 2              | 3              | 4              | 5              | 6              | 7              | 8              | 9              | **The gap with cRT** |
> > |-------------------------------------|----------------|----------------|----------------|----------------|----------------|----------------|----------------|----------------|----------------|---------------------------|
> > | cRT Shi et al. (2021)$^\ast$                   | 72.28          | 69.58          | 65.16          | 61.41          | 58.83          | 55.87          | 53.28          | 51.38          | 49.51        | -              |
> > | TOPIC Cheraghian et al. (2021)                 | 64.10          | 55.88          | 47.07          | 45.16          | 40.11          | 36.38          | 33.96          | 31.55          | 29.37        | -20.14         |
> > | CEC Zhang et al. (2021)                        | 73.07          | 68.88          | 65.26          | 61.19          | 58.09          | 55.57          | 53.22          | 51.34          | 49.14        | -0.37          |
> > | F2M Shi et al. (2021)                          | 71.45          | 68.10          | 64.43          | 60.80          | 57.76          | 55.26          | 53.53          | 51.57          | 49.35        | -0.16          |
> > | **LIMIT+Transformer** Zhou et al. (2022)                       | 73.81          | 72.09          | 67.87          | 63.89          | 60.70          | 57.77          | 55.67          | 53.52          | 51.23        | +1.72          |
> > | MetaFSCIL  Chi et al. (2022)                   | 74.50          | 70.10          | 66.84          | 62.77          | 59.48          | 56.52          | 54.36          | 52.56          | 49.97        | +0.46          |
> > | **ALICE+Data Aug** Peng et al. (2022)                       | 79.00          | 70.50          | 67.10          | 63.40          | 61.20          | 59.20          | 58.10          | 56.30          | 54.10        | +4.59          |
> > | Entropy-Reg  Liu et al. (2022)                 | 74.40          | 70.20          | 66.54          | 62.51          | 59.71          | 56.58          | 54.52          | 52.39          | 50.14        | +0.63          |
> > | C-FSCIL Hersche et al. (2022)                  | 77.50          | 72.45          | 67.94          | 63.80          | 60.24          | 57.34          | 54.61          | 52.41          | 50.23        | +0.72          |
> > | FSLL Mazumder et al. (2021)                    | 64.10          | 55.85          | 51.71          | 48.59          | 45.34          | 43.25          | 41.52          | 39.81          | 38.16        | -11.35         |
> > | FSLL+SS Mazumder et al. (2021)                 | 66.76          | 55.52          | 52.20          | 49.17          | 46.23          | 44.64          | 43.07          | 41.20          | 39.57        | -9.94          |
> > | HardNet (best)                                 | 78.35          | 74.12          | 70.13          | 65.88          | 62.74          | 59.56          | 57.98          | 56.31          | 54.32        | +4.81          |
> > | **SoftNet (best)**                                 | **79.88**      | **75.54**      | **71.64**      | **67.47**      | **64.45**      | **61.09**      | **59.07**      | **57.29**      | **55.33**    | **+5.82**      |

---

> > ### Author Response · Authors · 2022-11-16
> > **Additional Comparisons with Current SOTAs.**
> >
> > - Leveraged **only by regularized backbone ResNet18, SoftNet** outperformed all existing current works on CIFAR100 as shown in Table 1. of CIFAR100. On miniImageNet (Table 2.) and CUB-200-201 (Table 3.), the performances of SoftNet were comparable with those of ALICE and LIMIT, considering that **ALICE used class/data augmentations** and **LIMIT added an extra multi-head attention layer**.
> >
> > - In conclusion, given that **ALICE was recently published (Oct 25, 2022)**, we show that **SoftNet has achieved the best performance on CIFAR100 and miniImageNet, and is on par with LIMIT** at this ICLR submission time (Sep 28, 2022).

---

> ### Author Response · Authors · 2022-12-02
> **A Gentle Reminder**
>
> Dear Reviewer 5XQs,
>
> We sincerely appreciate your time and effort in reviewing our paper. During the discussion period, we have made every effort to faithfully address all your comments in the responses, by providing additional experimental results you requested, and revising the paper. We strongly believe that the paper is now significantly strengthened thanks to your constructive comments. Thus, we politely ask you to go over our responses and reconsider your rating on our work. Please let us know if you have any further questions.
>
> Best regards, Authors

---

> > ### Comment · Reviewer_5XQs · 2022-12-07
> > **Thanks for the response**
> >
> > Thanks for the detailed results and response. I still have the following concerns.
> >
> > 1.
> > I agree that ALICE (ECCV'22) is recently published and you do not have to compare with it. But CEC (CVPR'21) is am important baseline in few-shot class incremental learning. I do not think it could be overlooked in your original submission. What is the difference between SoftNet (Best) in the response and the SoftNet in your original submission that has much worse performance than SoftNet(Best)? As far as I am concerned, the SoftNet of your original submission still cannot surpass CEC in Tabel 1. By the way, I think these experimental works should have been done in your original submission.
> >
> > 2.
> > I appreciate the comparison between SoftNet and AANet offered in the response. But I think the model structure and the technical method have only minor difference, e.g., two residual blocks v.s. one residual block. I still believe that the key motivation and novelty have been covered by AANet (CVPR'21). It does not convince me of the significance of this work.
> >
> > So, I would like to keep the original rating.

---

> > > ### Author Response · Authors · 2022-12-07
> > > **We have already clearly resolved all of your concerns in our initial responses (Please see Table1,2,3 for CEC comparison and  Table 4,5 for our contributions compared to AANet)**
> > >
> > > Thanks for responding to our rebuttals and agreeing to exclude ECCV2022 papers (ALICE) in ICLR2023. **In rebuttal periods**, as the reviewer 5XQs suggested, we **addressed all concerns about the extensive comparisons of SoftNet with others** and **the main difference between SoftNet and AANet.**
> > >
> > > - We agree that CEC (CVPR'21) is an essential baseline in FSCIL. However, unlike other works (TOPIC, CEC, etc.) evaluated under a single fixed split, **we follow F2M experiment settings by evaluating all methods for random splits and ten-time evaluation.** We added all additional results in Appendix, including TOPIC split results.
> > >
> > > - Regarding the concern about the difference between SoftNet (Best) in the response and the SoftNet in our original submission, **the original SoftNet performances are based on a random split and 10-time evaluations as F2M does**, whereas SoftNet (Best) in the response refers to the results obtained from evaluation under a single fixed split. Furthermore, we demonstrated that **regularized sparse solution tends to show better generalization in FSCIL setting.** In particular, SoftNet (Best) outperformed CEC baselines, which use dense parameters as shown in **Tables 1,2, and 3.** We specify these results in the Appendix.
> > >
> > > - And regarding the main difference between SoftNet and AANet (CVPR'21), as we depicted the difference in **Table 4 in our responses**, SoftNet tries to find sparse solutions (representation) within a single ResNet. On the other hand, AANet learns a linear combination of multi-representations acquired by two dense ResNet. In summary, **AANet does not consider sparse solutions but is over-parameterized in FSCIL task**.
> > >
> > > - We appreciate the recommended works and papers to strengthen our novelty. Please let us know if you have any further questions.
> > >
> > > Best regards, Authors.

---

> > > ### Author Response · Authors · 2022-12-07
> > > **The responses to CEC (CVPR'21) baseline (TOPIC split)**
> > >
> > > We agree that CEC (CVPR'21) is an essential baseline in FSCIL. However, unlike other works **(TOPIC, CEC, etc.) evaluated under a single fixed split**, **we followed F2M experiment settings by evaluating all methods for random splits and ten-time evaluation** to show the effectiveness of our SoftNet. As the reviewer suggested, we added all extensive results in the Appendix, including TOPIC split results.

---

> > > ### Author Response · Authors · 2022-12-07
> > > **The responses to the difference between SoftNet (Best) in the response and SoftNet in our original submission.**
> > >
> > > - **The original SoftNet** performances are based **on a random split and 10-time evaluations as F2M conducted**, whereas **SoftNet (Best) in the response** refers to the results obtained from evaluation **under a single fixed split**.
> > >
> > > - Furthermore, we demonstrated that **regularized sparse solution (SoftNet) tends to show better generalization in the FSCIL setting.** In particular, **SoftNet (Best) outperformed CEC baselines, which use dense parameters**, as shown in Tables 1,2, and 3. We specify these results in the Appendix.

---

> > > ### Author Response · Authors · 2022-12-07
> > > **The responses to the difference between AANet and SoftNet.**
> > >
> > > - As we depicted the difference in **Table 4 in our responses**, **SoftNet tries to find sparse solutions (representation) within a single ResNet.** On the other hand, **AANet learns a linear combination of multi-representations acquired by two dense ResNet.**
> > >
> > > - In summary, **AANet does not consider sparse solutions but is over-parameterized in the FSCIL task.**

---

### Official Review · Reviewer_ZoLy · 2022-10-24

**Confidence:** 3
**Correctness:** 3
**Technical Novelty And Significance:** 3
**Empirical Novelty And Significance:** 3
**Recommendation:** 6

**Clarity, Quality, Novelty And Reproducibility:**

The work is well written and easy to follow. The idea is quite interesting and the methodology is in general clarified. The authors did not mention if they would release the source project on public.

**Strength And Weaknesses:**

Strengths:

S1) Few-shot Class Incremental Learning is an important and active topic in the community, which is addressed in this work.

S2) The paper is generally well written and easy to follow.

S3) The idea of jointly learns the model weights and adaptive soft masks to minimize catastrophic forgetting and to avoid overfitting novel few samples is interesting and makes sense.

S4) The method significantly surpasses many State-Of-The-Art methods under various variants on the benchmarks.

Weakness:

W1) Some of the details are not fully discussed. For example in section 3.3, why the update scheme of $\theta$ can effectively regularize the weights of the subnetworks for incremental learning.

W2) Though the effectiveness of SoftNet is verified via extensive experiments. These experiments are conducted only on two datasets (CIFAR-100 and miniImageNet), which seem not adequate enough to show the robustness of the propsed method. Results on other popular benchmark datasets such as CUB200 may add to the convincingness of the SoftNet.

**Summary Of The Paper:**

Inspired by Regularized Lottery Ticket Hypothesis, which hypothesizes that smooth subnetworks exist within a dense network, this paper propose Soft-SubNetworks (SoftNet), an incremental learning strategy that preserves the learned class knowledge and learns the newer ones. The SoftNet jointly learns the model weights and adaptive soft masks to minimize catastrophic forgetting and to avoid overfitting novel few samples in Few Shot Class-Incremental Learning. Experiments are conducted on the CIFAR-100 and miniImageNet datasets under various settings.

**Summary Of The Review:**

The paper tackles an important problem, proposing an incremental learning strategy that preserves the learned class knowledge and learns the newer ones.  Experiments are comprehensive and seem to show improved performance in two common benchmarks.

I have carefully reviewed all the comments and responses. Though a reviewer show concerns about the inadequate comparison with SOTAs, I still consider that this paper provides an interesting and effective idea. I would like to keep my original rating.

---

> ### Author Response · Authors · 2022-11-12
> **Outline of our responses**
>
> Dear Reviewer ZoLy,
>
> Thank reviewer ZoLy for constructive feedback. We will address all concerns about regularizing SoftNet parameters, extensive results, and Public code. After completing the additional experiments, we would like to prepare our responses to the reviewer's comments as follows:
>
> - Regularizing SoftNet parameters: we will explain the motivation for regularizing parameters in the processing of training.
>
> - Extensive experiments: as reviewer 5XQs suggested, we will report the comparisons SoftNet with seven different current methods on CIFAR100, miniImageNet, and CUB200.
>
> - Official codes: we will release our public code, including additional experimental results, soon.

---

> > ### Author Response · Authors · 2022-11-16
> > **Regularizing SoftNet parameters and Extensive Experiments**
> >
> > ### Regularizing SoftNet parameters.
> >
> > - We argue that there are two main reasons that the update scheme of SoftNet effectively regularizes subnetwork parameters $m_{soft} \odot {\theta}$:
> >
> >     - First of all, SoftNet obtains minor subnetwork $m_{minor}$ sampled from uniform distribution $U(0,1)$ at the base training session independent of the major subnetwork $m_{major}$ that learns the base session knowledge.
> >
> >     - Using the fact that $m_{minor} < {1}$, it would be highly useful in the exploration of subnetworks within the incoming few samples in the FSCIL setting.
> >
> > ### Extensive experiments.
> >
> > - In addition to two datasets, we have also conducted extensive experiments on the CUB200 dataset to compare SoftNet with existing methods in the Appendix.
> >
> > - As the reviewer 5XQs suggested, we included additional experiments on different splits.
> >
> > ### Official codes.
> > - We will publish our code online, including additional experiments, to justify the correctness and reproducibility of our work.

---

### Official Review · Reviewer_5Go7 · 2022-10-26

**Confidence:** 3
**Clarity, Quality, Novelty And Reproducibility:** see strength
**Correctness:** 3
**Technical Novelty And Significance:** 4
**Empirical Novelty And Significance:** Not applicable
**Recommendation:** 8

**Strength And Weaknesses:**

Strengths:
1.	Borrowing idea from regularized lottery ticket hypothesis to incremental learning is a novel and ingenious idea for its both maintaining performance in subnetwork and solving overfitting.
2.	The paper is well written for its clear logic, figures, equations, and organization.
3.	The outstanding performance of SoftNet is convincing.
Weaknesses:
1.	The choice of top-c weights is handcraft without theory support. Just comparing results on overall accuracy is not enough. The authors can refer to the works in subnetwork to refine their analysis.
2.	The random choice of parameter follows uniform distribution, but condensation argues that there are always some weights much important than the others. The authors should explain why they use uniform distribution or the future improvement in this prior.
3.	Is there any possibility of extending SoftNet to the other kind of incremental learning? Or is the SoftNet limited to only class-incremental learning?


**Summary Of The Paper:**

The paper proposes a brand-new few-shot class-incremental learning method called soft-subNetworks (SoftNet). The two main problems in few-shot class-incremental learning contains catastrophic forgetting and overfitting. The former one asks the new session training not to interfering the former session and the latter one asks only updating a few parameters irrelevant to previous tasks. Inspired by Regularized Lottery Ticket Hypothesis, the paper points out that the subnet can perform on-par or better than the whole network. Taking usage of this hypothesis, the authors split the network into a major subnetwork and a minor one. In the base classes session, soft-subnetwork parameters and weight score are learned. In the incremental learning session, minor parameters of the subnetwork are updated. The SoftNet surpasses the state-of-the-art baselines over datasets.

**Summary Of The Review:**

The paper outstands for its novel and the ingenious subnetwork training strategy. Also, the surpassing experiment results on benchmarks convince its idea. However, the theoretical analysis is quite week so that the readers cannot get the clear knowledge of the mechanism of subnetwork. The theoretical analysis should include how the split of major and minor help the forgetting and overfitting. Overall, the advantage outweighs its disadvantage so it can be recommended to the community.

---

> ### Author Response · Authors · 2022-11-12
> **Outline of our responses**
>
> Dear Reviewer 5Go7,
>
> Thank reviewer 5Go7 for constructive feedback. We will address all concerns about subnetwork references, the motivation for scaling minor weights using a uniform distribution, future improvements, and the broad impact on other fields. After completing the additional experiments, we would like to prepare our responses to the reviewer's comments as follows:
>
> - Subnetwork references: we will prepare subnetwork references to support our analysis of SoftNet.
>
> - The motivation for scaling minor weights using a uniform distribution: we will explain the motivation of SoftNet in terms of SoftNet Convergence with comparisons with loss landscapes (DenseNet, HardNet, and SoftNet) and scaling minor parameters (regularizing effect).
>
> - Future improvements: we will consider class-wise soft networks as a future direction to estimate an individual class distribution of given data.
>
> - The broad impact on other fields: SoftNet could be broadly extended to various works such as imbalanced dataset learning, debiased, neural network calibration, domain adaptation, transfer learning, and continual learning if we acquire an optimal class-wise or domain SoftNets.

---

> > ### Author Response · Authors · 2022-11-16
> > **The choice of top-c weights with subnetwork references.**
> >
> > - As the reviewer, 5Go7 suggested, we have tried to find theory supports for choosing top-c weights to strengthen our results and analysis from prior works.
> >
> > - In the related work section, we would also add that prior works have explored how we select top-$c\%$ weights from a dense network to acquire an optimal subnetwork.
> >
> > - Notably, random weighted networks (Ramanujan et al., 2020), building their work based on lottery tickets (Zhou et al., 2019), assigned a score $s$ to each connection of the network $\theta$ before it proceeds with the edge-popup algorithm (update scores, $s$) in finding the subnetwork within a randomly initialized network that belongs to the top-$c\%$ of $s$.
> >
> > - Naturally, this involves exploring data-relevant parameters, which proved to be useful in discovering optimal subnetworks (Wortsman et al., 2019; Ramanujan et al, 2020).
> >
> > - This method has been explored in various learning applications, from selecting winning subnetworks for forget-free continual learning and finding optimal subnetworks for few-shot meta-Learning (Chijiwa et al., 2022) to, most importantly, obtaining a soft subnetwork (SoftNet).
> >
> > - References
> >     - Vivek Ramanujan et al., What’s hidden in a randomly weighted neural network-CVPR2020
> >     - Hattie Zhou et al., HaDeconstructing lottery tickets: Zeros, signs, and the supermask - NeuralPS2019
> >     - Mitchell Wortsman et al., Discovering neural wiring-NeuralPS2019
> >     - Daiki Chijiwa et al., Meta-ticket: Finding optimal subnetworks for few-shot learning within randomly initialized neural network - NeuralPS2022

---

> > ### Author Response · Authors · 2022-11-16
> > **Uniform selection of random minor parameters and The theoretical analysis**
> >
> > ### Uniform selection of random minor parameters.
> >
> > - The motivation for using uniform distribution: we provide two motivations for randomly selecting minor parameters $m_{minor}$ based on uniform distribution $U(0,1)$.
> >
> >     - (1) During training, sampling $m_{minor}$ from uniform distribution $U(0,1)$ is equivalent to injecting noise into the binary mask $m_{major}$. This binary mask scaling of minor parameters can be seen as a way to achieve global minima. In the Appendix, we add a mathematical background based on the convergence theorem and compare loss landscapes between DenseNet, HardNet, and SoftNet.
> >
> >     - (2) More importantly, this binary mask scaling can be seen as a way of regularizing model parameters to prevent overfitting, leading to more diverse features. In base session training, the major mask tries to acquire a regularized subnetwork, while the minor tries to help explore the flat minima.
> >
> > ###  The theoretical analysis should include how the split of major and minor help the forgetting and overffitting.
> > - During training, the minor mask ${m_{minor}}$ sampled through uniform distribution corresponds to the random exploration of the weights. It is equivalent to inducing noise to minor parameters to achieve flatter global minima and avoid overfitting. In the appendix, we prove that HardNet and SoftNet achieve convergence and compare the loss landscape of DenseNet with both aforementioned subnetworks.

---

> > ### Author Response · Authors · 2022-11-16
> > **The future improvement and The extension of SoftNet**
> >
> >
> > ### The extension of SoftNet to other kinds of incremental learning.
> > - Selecting $m_{minor}$ via uniform distribution does not guarantee to obtain an optimum subnetwork.
> > - To prove its optimality, we must guarantee the existence of a soft subnetwork within a dense network representing data distribution.
> > - As a future work, one possible approach is to acquire class-wise SoftNet, which corresponds to the data distribution of a particular class.
> > - Next, we superposition these class-wise SoftNets to obtain an optimal SoftNet representing the whole data distribution.
> > - To verify its optimality, one can compare it with the randomly initialized SoftNet.
> >
> > ### The extension of SoftNet to other kinds of incremental learning.
> > - SoftNet is highly applicable to real-world problems, where we learn continuously from newly arriving inputs with random data sizes. In this setting, the extended SoftNet should be able to both expand the task-relevant new parameters and re-select the subset of previously learned parameters.
> >
> > - Regarding applicable tasks, we expect that SoftNet can be adapted to various tasks not limited to classification on imbalance dataset, debiasing, neural network calibration, domain adaptation, transfer learning, and continual learning by acquiring optimal class-wise or domain SoftNets.

---

### Decision · Program_Chairs · 2023-01-20

**Decision:**

Accept: poster

**Justification For Why Not Higher Score:**

Two reviewers recommended acceptance, and one recommended weak rejection (less negative after rebuttal). Unfortunately, we did not reach a consensus. The paper can benefit from implementing the feedback from the reviewers especially related work coverage. The authors partially address these concerns in the rebuttal.

**Justification For Why Not Lower Score:**

Two reviewers recommended acceptances, and one recommended week rejection. Despite that, there is no consensus, the reviewers agree on the significance of the results. On balance, the paper will be a valuable addition to the ICLR program.

**Metareview: Summary, Strengths And Weaknesses:**

The paper proposes a few-shot incremental learning approach, referred to as Soft-SubNetworks (SoftNet). The approach is inspired by
the Regularized Lottery Ticket Hypothesis, which states that competitive smooth (non-binary) subnetworks exist within a dense network,


Strengths
------------

- Reviewer 5Go7, Reviewer ZoLy: Borrowing idea from regularized lottery ticket hypothesis to incremental learning is a novel and ingenious idea for its both maintaining performance in subnetwork and solving overfitting
- Reviewer 5Go7, Reviewer ZoLy, Reviewer 5XQs: The paper is well written its clear logic, figures, equations, and organization.
- Reviewer 5Go7, Reviewer ZoLy: The outstanding performance of SoftNet is convincing
- Reviewer 5XQs: The method is simple and easy to follow.


Weaknesses
---------------

- Reviewer ZoLy, Reviewer 5Go7: Some of the details are not fully discussed. For example in section 3.3, why the update scheme or can effectively regularize the weights of the subnetworks for incremental learning.
- Reviewer ZoLy: Though the effectiveness of SoftNet is verified via extensive experiments. These experiments are conducted only on two datasets (CIFAR-100 and miniImageNet), which seem not adequate enough to show the robustness of the propsed method. Results on other popular benchmark datasets, such as CUB200, may add to the convincingness of the SoftNet.
-Lack of related work coverage and discussion  (e.g., below by   Reviewer 5XQs)
Few-shot incremental learning with continually evolved classifiers. (CVPR ‘21)
Few-shot class incremental learning by sampling multi-phase tasks. (TPAMI)
Subspace regularizers for few-shot class incremental learning. (ICLR ‘22)
Metafscil: A meta-learning approach for few-shot class incremental learning. (CVPR ‘22)
Constrained few-shot class-incremental learning. (CVPR ‘22)
Few-shot class incremental learning via entropy-regularized data-free replay. (ECCV ‘22)
Few-shot class-incremental learning from an open-set perspective. (ECCV ‘22)


In conclusion, two reviewers recommended acceptance, and one recommended weak rejection (less negative after rebuttal). . On balance, the AC finds the paper could be accepted after the improvement in the rebuttal. However, the authors should attend to the main points in the review when preparing a final version. ,


**Note From Pc:**

if the above contains the word "oral" or "spotlight" please see: "oral" presentation means -> notable-top-5% and "spotlight" means -> notable-top-25%. As stated in our emails, we are disassociating presentation type from AC recommendations

**Summary Of Ac-Reviewer Meeting:**

N/A.